# Attention network modulation via tRNS correlates with attention gain

**Federica Contò[1,2]\*, Grace Edwards[1,3], Sarah Tyler[1,4], Danielle Parrott[1,2], Emily Grossman[5], Lorella Battelli[1,2,3,6]**

[1]Center for Neuroscience and Cognitive Systems@UniTn, Istituto Italiano di Tecnologia, Rovereto, Italy; [2]Center for Mind/Brain Sciences, University of Trento, Rovereto, Italy; [3]Department of Psychology, Harvard University, Cambridge, United States; [4]Butte College, Oroville, United States; [5]Department of Cognitive Sciences, University of California, Irvine, Irvine, United States; [6]Department of Neurology, Berenson-Allen Center for Noninvasive Brain Stimulation, Beth Israel, Deaconess Medical Center, Harvard Medical School, Boston, United States

**Abstract** Transcranial random noise stimulation (tRNS) can enhance vision in the healthy and diseased brain. Yet, the impact of multi-day tRNS on large-scale cortical networks is still unknown. We investigated the impact of tRNS coupled with behavioral training on resting-state functional connectivity and attention. We trained human subjects for 4 consecutive days on two attention tasks, while receiving tRNS over the intraparietal sulci, the middle temporal areas, or Sham stimulation. We measured resting-state functional connectivity of nodes of the dorsal and ventral attention network (DVAN) before and after training. We found a strong behavioral improvement and increased connectivity within the DVAN after parietal stimulation only. Crucially, behavioral improvement positively correlated with connectivity measures. We conclude changes in connectivity are a marker for the enduring effect of tRNS upon behavior. Our results suggest that tRNS has strong potential to augment cognitive capacity in healthy individuals and promote recovery in the neurological population.

**\*For correspondence:**
Federica.Conto@iit.it

**Competing interest:** The authors declare that no competing interests exist.

## Editor's evaluation

Conto et al. applied transcranial random noise stimulation to the parietal cortex during performance of an attention task, examining if multiple sessions would result in a cumulative increase in performance. The results showed that tRNS to parietal, but not a ventral-lateral temporal site, increased performance on an orientation discrimination task, but not a temporal order judgement task. Using resting-state functional magnetic resonance imaging and a functional connectivity (FC) analysis, tRNS was shown to increase FC within the dorsal and ventral attention network. Overall, an increase in FC within the DVAN positively correlated with changes in behavioral performance (collapsing across all stimulation conditions). Overall, the experimental design and results are compelling. These findings further our understanding of how combining non-invasive brain stimulation with behavioral training can lead to enhanced neural plasticity that supports more effective learning.

## Introduction

Vision and attention are the primary sensory and cognitive modalities through which humans interact with the environment. Augmentation of visuoperceptual function through training, termed perceptual learning (PL), is crucial for clinical and aging populations that benefit from therapeutic interventions to recover and retain perceptual function. PL is also a pivotal tool for investigating neural plasticity

(*Gilbert, 1994*; *Polat et al., 2004*; *Shibata et al., 2016*; *Deveau et al., 2014*; *Sterkin et al., 2018*), and has been linked to localized neural changes in sensory cortex and long-range sensory-attention network reorganization (*Gilbert et al., 2001*; *Li et al., 2004*; *Sasaki et al., 2010*; *Fahle, 2004*). Thus, PL has potential as an approach to reveal underlying cortical reorganization that promotes sustained improvements in perceptual function.

Although promising, PL approaches have been limited in their applicability, due in part to the long and intensive training protocols which largely impede their use (*Dosher and Lu, 2017*; *Das et al., 2014*; *Li et al., 2004*; *Huang et al., 2012*; *Zhang et al., 2010a*; *Zhang et al., 2010b*; *Dosher and Lu, 2005*). Currently lacking are PL approaches to increase the rate of learning, valuable for populations with limited abilities to engage in demanding perceptual tasks. A notable exception is 'cross-tasks' training, in which learning protocols are significantly shortened by adopting a training procedure on two different tasks with the same stimuli simultaneously (*Szpiro et al., 2014*; *Wright et al., 2010*).

More recently, transcranial electrical stimulation (tES) has been proposed as a tool to facilitate cortical plasticity, particularly when coupled with behavioral training (*Edwards et al., 2019*). When delivered using a transcranial random noise protocol (tRNS), tES can modulate visual cortex excitability (*Herpich et al., 2018*), promote visual functions in healthy subjects (*Fertonani et al., 2011*; *Pirulli et al., 2013*; *Tyler et al., 2018*), and facilitate recovery of visual dysfunctions in disease (*Campana et al., 2014*; *Camilleri et al., 2016*; *Herpich et al., 2019*). Moreover, recent studies show PL coupled with tES can successfully boost the effects of training by increasing the rate of learning across sessions (*Fertonani et al., 2011*), particularly when applied during task training (*Pirulli et al., 2013*), with the subsequent observed behavioral improvements persisting for durations well beyond those obtained through PL training alone (*Cappelletti et al., 2013*; *Herpich et al., 2019*).

Studies on the physiological effects of tRNS on brain dynamics are not yet clear on the mechanisms by which training is facilitated. The leading hypothesis derived from animal studies is that tRNS has a stochastic resonance effect upon neurons such that the injected subthreshold stimulation both reso-nates with ongoing task-related oscillatory activity and reduces endogenous noise, and thus, when applied at optimal levels, enhances the signal-to-noise of task-related spike firing (*Antal and Herr-mann, 2016*; *Liu et al., 2018*; *Polanía et al., 2018*). Consistent with the network oscillation compo-nent of resonant theory, extant studies on humans show that focal tES of a node within a network can cause a cascade of functional changes selectively spreading within that network (*Nitsche and Paulus, 2000*; *Polanía et al., 2012*; *Antal et al., 2011*; *Krause et al., 2017*). Modeling studies have likewise shown changes in network dynamics within the directly stimulated area and across distal but connected cortical areas (*Kunze et al., 2016*; *Polanía et al., 2011*; *Sehm et al., 2012*). Given that many behavioral manifestations of neurological and psychiatric disorders are the consequence of altered brain network connectivity (*Buckner et al., 2005*; *Fox et al., 2014*), the network-wide impact of tES could have wide clinical applications.

The present study uses a combination of tRNS and fMRI, in conjunction with cross-training on a PL task, to investigate large-scale cortical dynamics when coupled with behavioral training. Specifically, we implemented a 4-day attention training protocol on two randomly interleaved PL tasks: orienta-tion discrimination (OD) and temporal order judgments (TOJs). This training was coupled with tRNS to examine if tRNS to the parietal lobe would modulate the dorsal and ventral attention network (DVAN), and subsequently increase attention. The target of the parietal stimulation was the intrapa-rietal sulcus (IPS), a crucial node of spatial attention (*Vossel et al., 2014*; *Battelli et al., 2017*; *Plow et al., 2014*; *Leitão et al., 2015*; *Battelli et al., 2009*). The attention network is often divided into the dorsal and the ventral attention sub-networks (DAN and VAN, or DVAN; *Corbetta and Shulman, 2002*; *Fox et al., 2006*). However, we are interested in the whole network due to the continuous interplay between networks, which enables the dynamic control of the different attentional processes (*Vossel et al., 2014*; *Macaluso and Driver, 2005*; *Sani et al., 2019*). Moreover, to control for network-specific effects of stimulation, we also measured whether tRNS and training affected dynamics within the default mode network (DMN), typically active during passive or resting conditions (*Raichle et al., 2001*; *Buckner and DiNicola, 2019*). Among the nodes of the DAN, the human middle temporal complex (hMT+) has been directly implicated in visual temporal encoding (*Salvioni et al., 2013*), a cognitive process involved by our tasks. Thus, hMT+ was included as a second stimulation site to evaluate the efficacy of tRNS on perceptual processes, and to serve as an active control to determine if attention modulation was specific to parietal stimulation. We also sought to understand whether

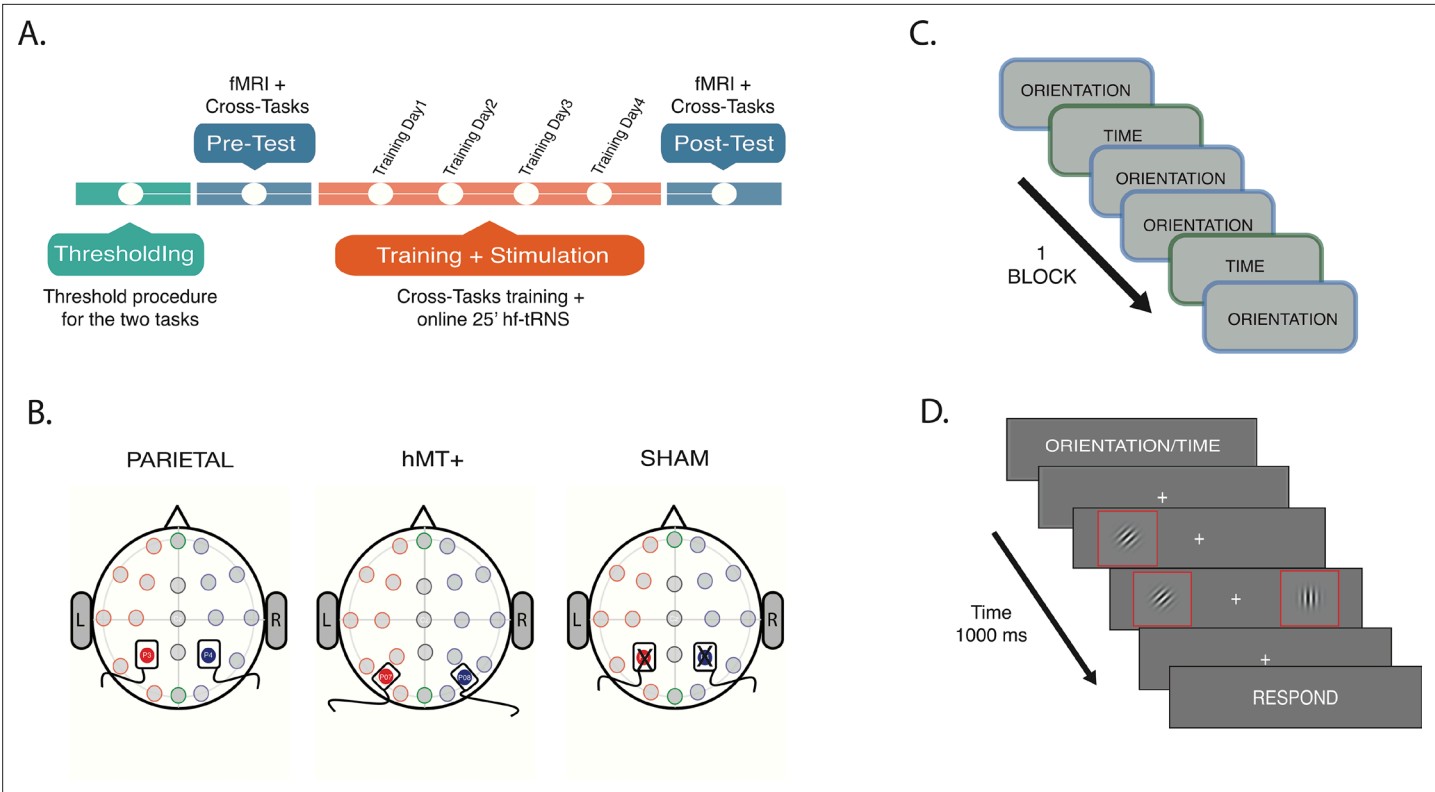

**Figure 1.** Procedure and stimulation sites. (**A**) Experiment timeline. Thresholds for OD and TOJ tasks were performed at the beginning of the multi-session experimental procedure (green section). On the pre-test session, subjects were tested on the tasks while fMRI data were collected (blue section). A resting-state scan was also collected during this session. Next, subjects underwent behavioral training concurrently with tRNS for 25 min (Training Day 1, Training Day 2, Training Day 3, and Training Day 4, orange section). Post-test session was a repeat of the pre-test session (blue section). For detailed information on fMRI data acquisition order, see Materials and methods section. (**B**) Stimulation settings. Location of stimulation sites were localized using EEG 10/20 system. Saline-soaked electrodes was placed over P3 (red filled circle) and P4 (blue filled circle) for bilateral parietal and Sham stimulation, and over P07 (red filled circle) and P08 (blue filled circle) for hMT+ stimulation. (**C**) Intermixed trials sequence example: two types of attention tasks were randomly presented within each block, the OD and the TOJ task. (**D**). Example trial. The visual information was the same for TOJ and OD tasks. The cue at the beginning of the trial dictated which feature (time or orientation) should be attended. Red outline boxes are used to highlight the Gabors stimuli (black and white oriented bars) in the illustration but they were not present during the experiment. OD, orientation discrimination; TOJ, temporal order judgment.

neural modulation following tRNS coupled with training would lead to long-lasting behavioral improvements, and we tested it 24 hr following the last training session by examining if behavioral and neuromodulatory changes persisted. Lasting neural and behavioral change would suggest cortical plasticity that outlasts the end of stimulation.

## Results

### Learning-dependent changes in behavior

We examined the effect of tRNS delivered concurrently with the task in a 4-day PL protocol (*Figure 1A*, orange section) that embedded two attentional tasks: an OD task in which participants were asked to determine if the Gabors (disks with black and white stripes, *Figure 1D*) presented either side of a fixation cross had the 'same' or 'different' orientation, and a TOJ task in which participants were asked to determine if the Gabors' onsets were simultaneous (same) or asynchronous (different) (*Figure 1C and D*; for a detailed description of the experimental procedure, tasks, and the stimuli see '*Materials and methods*' section). These two attention tasks, well known to increase activity within attention-related cortical areas, were randomly presented within each block, a 'cross-tasks' procedure which has shown efficacy in supporting more learning than single-task training (*Szpiro et al., 2014*). While engaged in the attention tasks, participants received concurrent tRNS to cortical regions crucial hotspots of

visuospatial attention, bilateral IPS, and over hMT+ or Sham in control conditions (*Figure 1B*). Pre and post-measures of task performance, as well as resting state connectivity, were collected prior and subsequent to the multi-day training sessions. This allowed us to evaluate both the effect of stimulation combined with training on each task separately (*Figure 1A*, blue section) and the time course of learning across training and stimulation days (*Figure 1A*, orange section).

## Magnitude of learning
### Orientation discrimination
We first ensured that the three stimulation groups did not differ prior to stimulation by comparing their behavioral performance on the orientation task. A one-way analysis of variance (ANOVA) revealed no significant difference between groups prior to stimulation (F(2, 27)=0.575, p=0.569, $\eta_p^2$=0.041). We then examined the combination of stimulation and training on learning. This analysis revealed a main effect of session on OD performance (F(1, 27)=10.056, p=0.004, $\eta_p^2$=0.271), indicating that accuracy scores changed as a function of training. There was no significant main effect of stimulation condition alone on performance (F(2, 27)=0.273, p=0.763, $\eta_p^2$=0.020). Crucially, the analysis revealed a significant interaction between stimulation condition and session (F(2, 27)=11.257, p<0.001, $\eta_p^2$=0.455), indicating that learning from the pre- to post-test sessions varied significantly depending on the stimulation site.

To further investigate the significant interaction effect, we ran pairwise t-tests to compare pre- and post-test performance for each group, separately. We found that only the Parietal group showed significant improvement in orientation judgment accuracy between the pre- and post-test sessions (t(9)=−4.53, p=0.001, average improvement 29.8%, Cohen's d=−1.43), while there were no significant differences between pre- and post-test performance for the hMT+ group (t(9)=−0.608, p=0.558, average improvement 2.2%, Cohen's d=−0.19), nor for the Sham group (t(9)=0.602, p=0.562, average decrease −3.1%, Cohen's d=−0.19). Average improvements per condition are reported in *Figure 2A*. A one-way ANOVA on post-test performance (F(2, 20.593)=3.672, p=0.043, Brown-Forsythe corrected; $\eta_p^2$=0.214) and on performance improvements values (F(2, 27)=11.257, p<0.001; $\eta_p^2$=0.455) confirmed a significant difference between the three stimulation groups. Bonferroni post-hoc pairwise comparisons revealed a significant difference in performance improvement between Parietal and Sham conditions (p=0.003, Cohen's d=1.76), and between parietal and hMT+ (p≤0.001, Cohen's d=1.64), while hMT+ and Sham did not differ significantly (p=1.0, Cohen's d=0.38).

### Temporal order judgment
A one-way ANOVA on pre-test session data revealed no significant differences between groups at baseline (F(2, 27)=0.14, p = 0.867; $\eta_p^2$=0.010). We next investigated whether there was a significant difference between the pre- and post-test TOJ performance depending on stimulation condition. A mixed factors repeated measures ANOVA revealed no main effect of session (F(1, 27)=0.496, p=0.48; $\eta_p^2$=0.018) nor of condition (F(1, 27)=0.669, p=0.521; $\eta_p^2$=0.047) on performance. There was also no significant interaction between session and stimulation condition (F(2, 27)=0.589, p=0.562; $\eta_p^2$=0.042), which indicates that performance in the pre- and post-test session did not differ depending on stimulation condition.

A one-way ANOVA on calculated improvement scores (difference between pre- and post-test performance, *Figure 2B*) confirmed a non significant performance change between stimulation conditions (F(2, 27)=0.589, p=0.562; $\eta^2$=0.042). Participants in the parietal tRNS group performed on average 3.1% better following multi-session stimulation and training, while there was a slight decrease in performance for the hMT+ (–4.1%) and the Sham groups (–7.8%), *Figure 2B*.

## Time course of learning
### Orientation discrimination
We next analyzed performance during the 4-day training sessions. To measure changes in performance over time, we performed a mixed repeated measures ANOVA with stimulation site ($n = 3$) as between-subjects factor and training day ($n = 4$) as within-subjects factor. Mauchly's test indicated that the assumption of sphericity was violated ($\chi^2$ (5)=19.7, p=0.001); hence, the Greenhouse-Geisser correction was used to correct the degrees of freedom and to assess the significance of the corresponding p-values. We found a significant main effect of training days (F(1.97, 53.31)=8.89, p<0.001;

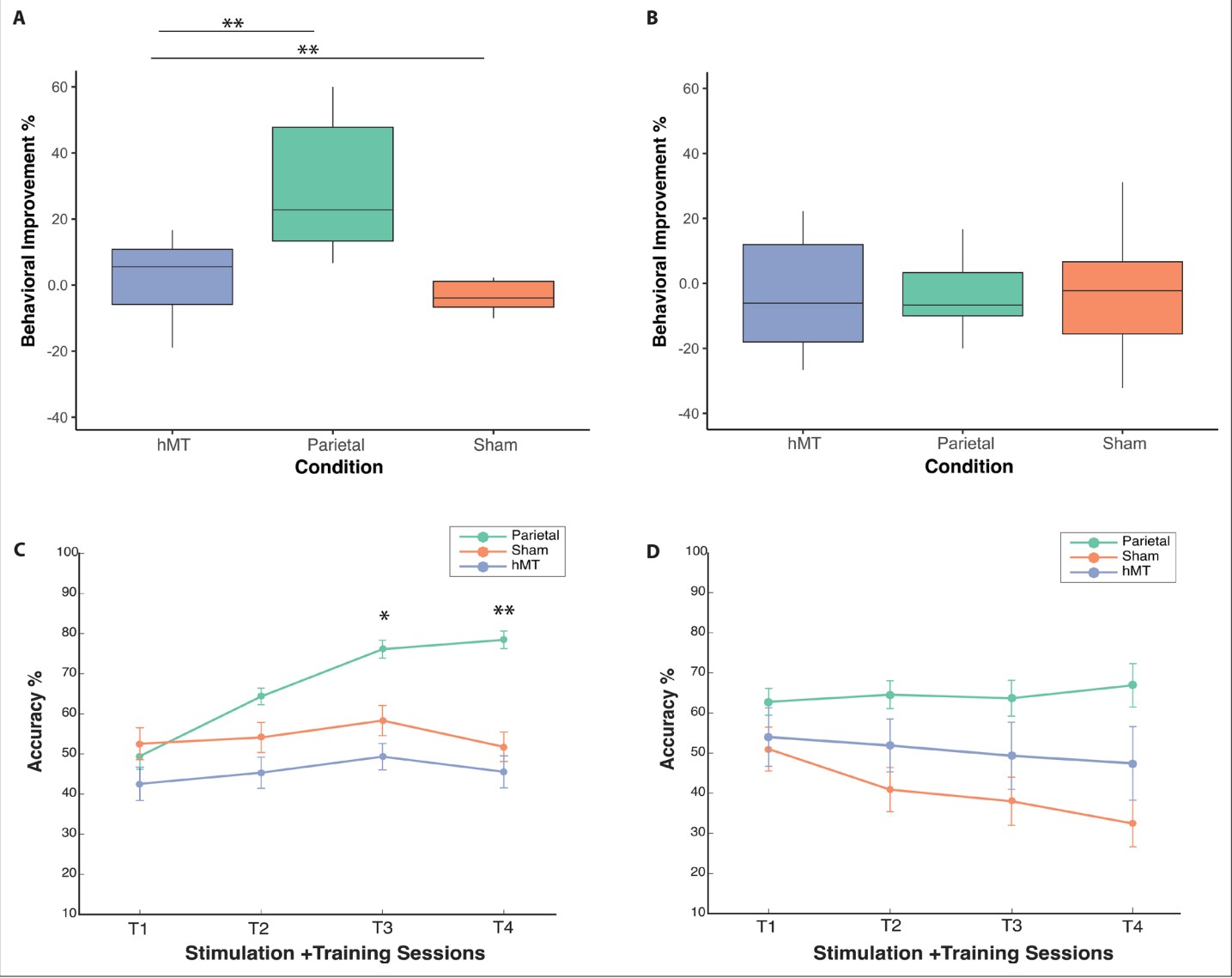

**Figure 2.** Behavioral performance on the OD and the TOJ task during and following stimulation and training. (**A**) Box plots for behavioral improvement by stimulation condition on the OD and (**B**) the TOJ task normalized to performance at baseline, prior to stimulation. The center line in the middle of the box is the median of each data distribution, while the box represents the interquartile range (IQR), with the lower quartile representing the 25th percentile (Q1) and the upper quartile representing the 75th percentile (Q3). (**C**) Change in accuracy across training days for all conditions on the OD and on the TOJ task (**D**) during training and stimulation sessions. Each line represents a different stimulation condition (green for parietal, orange for Sham, and blue for hMT+ group), error bars represent SEM. Asterisks indicate significant differences relative to the first training session (*<0.05, **<0.01). OD, orientation discrimination; TOJ, temporal order judgment.

$\eta_p^2$=0.248) but no main effect of stimulation condition (F(2, 27)=2.9, p=0.068; $\eta_p^2$=0.181). Importantly, the analysis revealed a significant two-way interaction between stimulation and training day on performance (F(3.94, 53.31)=4.37, p=0.004; $\eta_p^2$=0.245), indicating the effects of training on performance depended on stimulation condition.

To further investigate the time-dependent impact of stimulation on training performance and to analyze at what time point the groups began to diverge, we evaluated the effect of group on each training day separately. Levene's test was not significant for any training day (Training Day 1 p=0.778, Training Day 2 p=0.269, Training Day 3, p=0.742, and Training Day 4 p=0.297), indicating the assumption of homogeneity of variances was not violated. tRNS administered during training caused learning that was significantly different among stimulation groups from the third day of training (Training Day 3: F(2,27)=4.65, p=0.018; $\eta_p^2$=0.257) and remained sustained on the final day of training (Training

Day 4: F(2,27)=6.63, p=0.005; $\eta_p^2$=0.330). Post-hoc pairwise t-tests revealed the Parietal group performed significantly better than the hMT+ group on Training Day 3 (t(18)=3.6, p=0.00; Cohen's d=1.51) and approached significance when compared to Sham (t(18)=2.03, p=0.056, Cohen's d=0.91), while hMT+ and Sham did not differ significantly (t(18)=0.9, p=0.37, Cohen's d=−0.40). Similarly, on training Day 4, we found a significant difference in performance between the Parietal and the hMT+ groups (t(18)=3.6, p=0.002; Cohen's d=1.61) and between Parietal and Sham (t(18)=3.102, p=.006); Cohen's d=1.39, but not between hMT+ and Sham (t(18)=0.58, p=0.57; Cohen's d=−0.26).

Finally, we analyzed performance of the Parietal group only during training to investigate whether performance increased across sessions (Training Days 1, 2, 3, and 4). We found that performance on training Day 1 was significantly different from performance of training Day 2 (t(9)=−3.34, p=0.009; Cohen's d=−1.056), Day 3 (t(9)=−4.63, p=0.001; Cohen's d=−1.46), and Day 4 (t(9)=−5.05, p=0.001; Cohen's d=−1.59). These comparisons indicate that Parietal group performances on the OD task significantly increased on each stimulation training session relative to the first training session (*Figure 2C*).

## Temporal order judgment

We ran the same analysis on training data for the TOJ task. Mauchly's test for the mixed factor ANOVA indicated that the assumption of sphericity was violated ($\chi^2$ (5)=18.86, p=0.002); hence, the Greenhouse-Geisser correction was used to correct the degrees of freedom and to assess the significance of the corresponding p-values. There was no significant effect of training session on TOJ performance (F(1.8, 53.50)=2.707, p=0.076; $\eta_p^2$=0.091), but there was a significant main effect of stimulation condition (F(2, 27)=4.79, p=0.017; $\eta_p^2$=0.262). For the mean performance evaluated during the tRNS session, but notable not pre- and post-training (above), the Parietal stimulation group performance was significantly better than that of the Sham group (post hoc t-test: p=0.014; Bonferroni corrected; Cohen's d=1.84), but not the hMT+ group (post hoc t-test: p=0.256, Bonferroni corrected; Cohen's d=0.77), no significant difference was found also between Sham and hMT+ groups (post hoc t-test: p=0.618, Bonferroni corrected; Cohen's d=0.50). We note, however, that there was no interaction between stimulation and training session on performance (F(3.96, 53.506)=2.277, p=0.073; $\eta_p^2$=0.144), indicating that the tRNS did not preferentially benefit training for any stimulation group on this TOJ task (*Figure 2D*).

## Stimulation and learning-dependent changes in functional connectivity

We next examined the effect of multi-session hf-tRNS and training on resting-state functional connectivity (rs-FC) of the DVAN, the functional network targeted by stimulation and training protocol.

To analyze the effect of stimulation and training on rs-FC of the DVAN, we first calculated the mean FC per subject within the DVAN by averaging the FC scores of 45 connectivity values obtained from a matrix derived for each subject for all the region of interest (ROI) pairs of the DVAN (see Materials and methods for details), we then averaged across subjects in each group (*van den Heuvel et al., 2017*; *Nicolini et al., 2020*).

To ensure that the number of data points remaining after time series were cleaned from motion did not bias the r-to-z transformations, we performed statistical analysis to test whether the number of censored frames was equivalent across conditions. A two-way mixed repeated measures ANOVA was performed to evaluate whether the number of censored data frames was affected by stimulation conditions and time (fMRI session). There was homogeneity of variances on both sessions (p>0.05) as assessed by Levene's test. There was no statistically significant interaction between stimulation group and time on censored frames (F(2, 27)=1.604, p=0.220, $\eta_p^2$=0.034). Finally, we found no main effect of condition (F(2, 27)=0.038, p=0.963; $\eta_p^2$=0.003) nor session (F(1, 27)=0.619, p=0.438; $\eta_p^2$=0.007) indicating that the mean number of censored data frames was equivalent across conditions and sessions.

We compared FC prior to stimulation between groups and found no significant difference (F(2, 27)=0.300, p=0.743; $\eta_p^2$=0.22), indicating that FC did not differ between stimulation groups at baseline (pre-test session). A mixed repeated measures ANOVA testing the two factors of session (within-subjects factor, $n = 2$) and stimulation site (between-subjects factor, $n = 3$) on functional connectivity scores revealed no main effect of training on FC scores (F(1, 27)=0.001, p=0.986; $\eta_p^2$=0.00) and a main effect of stimulation site (F(2, 27)=4.064, p=0.029; $\eta_p^2$=0.231), indicating that the connectivity among the DVAN nodes changed as a function of where tRNS was applied. Importantly,

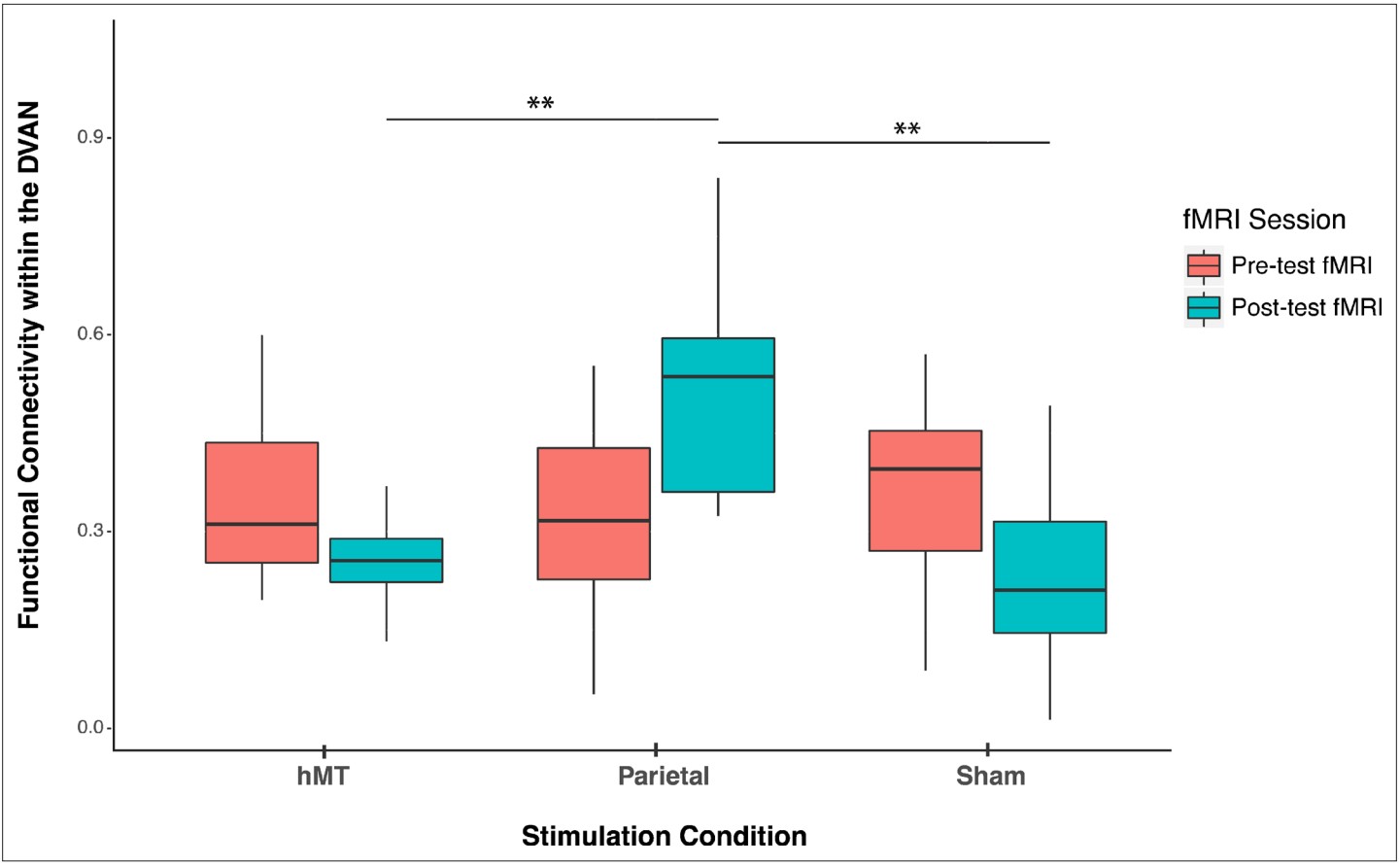

**Figure 3.** Pre- and post-stimulation functional connectivity (FC) changes. Box plots for overall mean FC of the dorsal and ventral attention network averaged per stimulation condition (Parietal, hMT+, and Sham) and per fMRI session (pre- and post-test sessions). Asterisks indicate significant difference (**<0.01).

the interaction between stimulation condition and session was found to be highly significant (F(2, 27)=11.507, p<0.001; $\eta_p^2$=0.460), indicating that the change in rs-FC scores differed depending on the stimulation group (**Figure 3**) and this change was persistent the day following the last combined training and stimulation session.

$$n = 3$$

To detect whether FC was impacted by stimulation condition, we compared the rs-FC of the three stimulation groups after stimulation and training. One-way ANOVA analysis on post-test session revealed a significant effect of stimulation condition on FC scores (F(2, 27)=11.677, p<0.001; $\eta_p^2$=0.464), which indicates that FC differs on this day between stimulation conditions (**Figure 3**). We then ran t-test comparisons on post-test FC scores to reveal simple effects between pairs of stimulation conditions. As depicted in **Figure 3**, we found that FC scores were significantly different in the Parietal group compared to the hMT+ (t(18)=3.863, p=0.002; Cohen's d=1.73) and to the Sham group (t(18)=3.64, p=0.001; Cohen's d=1.63), while FC did not differ significantly between Sham and hMT+ groups (t(18)=0.187, p=0.854; Cohen's d=0.08). We further investigated the effect of stimulation on FC by comparing pre- and post-test FC values within each stimulation condition. Pairwise t-test comparisons revealed that FC scores were significantly different between the pre- and post-test sessions in the Parietal (t(9)=−2.973, p=0.016; Cohen's d=−0.94) and the hMT+ groups (t(9)=−2.99, p=0.015; Cohen's d=0.95), while they were not significantly different in the Sham group (t(9)=2.217, p=0.054; Cohen's d=0.70). Rs-FC patterns increased within the main nodes of the DVAN after parietal stimulation only, while they decreased after hMT+ and Sham stimulation (**Figure 3**).

In a second set of analysis, we investigated the modulation of the DVAN by taking into consideration all ROIs pairs as single components of the network per subject (n=45 ROIs to ROIs pairs per

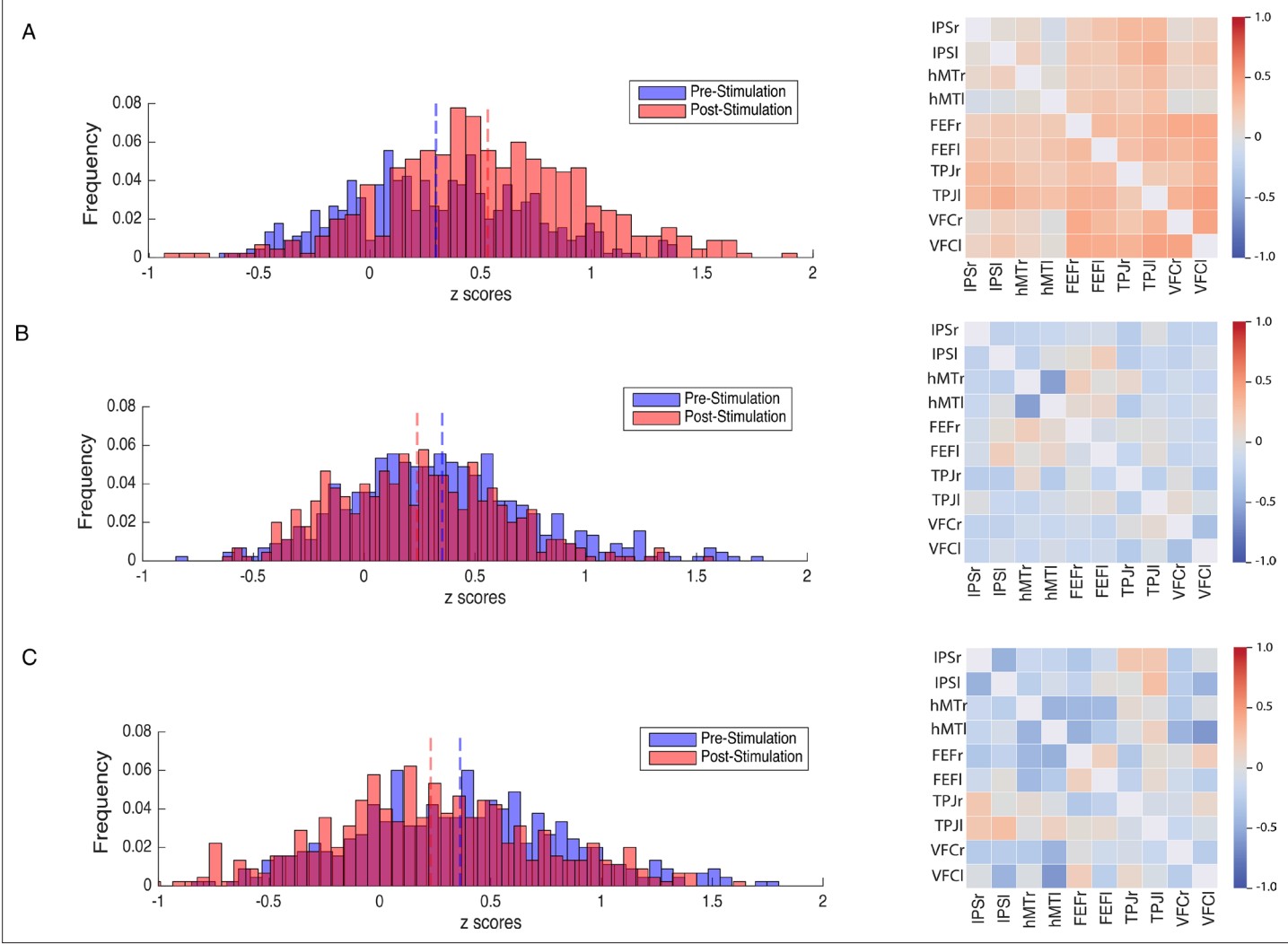

**Figure 4.** Modulation of resting-state functional connectivity (FC) of the DVAN. Left Panel: z-scores frequency distribution pre- (light blue bars) and post-test (light red bars) multi-session hf-tRNS coupled with training per each stimulation condition (A. Parietal, B. hMT+, C. Sham). Dark red indicates the overlapping distribution between pre- and post-test sessions. Dotted red and blue lines indicated the mean for each z-scores distribution. Right Panel: Correlation matrices represent the result of the computed difference between FC scores pre- and post-test sessions (Δ=FC(S2)−FC(S1)) depicted by condition (A. Parietal, B. hMT+, C. Sham). Each correlation difference between ROI pairs was first calculated at subject level and then averaged across subjects (detailed information on ROIs are reported in the Materials and methods section). Colors bars indicate the strength and direction of correlation change values for each regions' pair (red colors indicate higher connectivity; blue colors indicate lower connectivity). DVAN, dorsal and ventral attention network; ROI, region of interest; tRNS, transcranial random noise stimulation.

subject), instead of using one averaged correlation value for the whole network as in the previous analysis. As for the previous analysis, we performed a one-way ANOVA to compare FC between groups prior to stimulation and found no significant difference ($F(2, 1317.1)=1.556$, $p=0.211$, Brown-Forsythe p-value and degrees of freedom corrected; $\eta_p^2=0.002$), indicating that FC did not differ between stimulation groups at baseline (pre-test session). We next performed a mixed repeated measures ANOVA to test the effect of the stimulation site and session on all correlation values within the DVAN. This analysis revealed no main effect of session on FC scores ($F(1, 1347)=.002$, $p=0.964$; $\eta_p^2=0.00$), but a main effect of stimulation group ($F(1, 1347)=23.07$, $p\leq0.001$; $\eta_p^2=0.106$). Importantly, the interaction between stimulation group and session was found to be strongly significant ($F(2, 1347)=80.206$, $p<0.001$; $\eta_p^2=0.033$), which indicates a significant difference between rs-FC scores pre- and post-test sessions, depending on the stimulation condition (*Figure 4*).

We then compared the rs-FC on all single components (ROIs-to-ROIs pairs) of the post-test session by performing a one-way ANOVA. Because Levene's test indicated that the assumption of

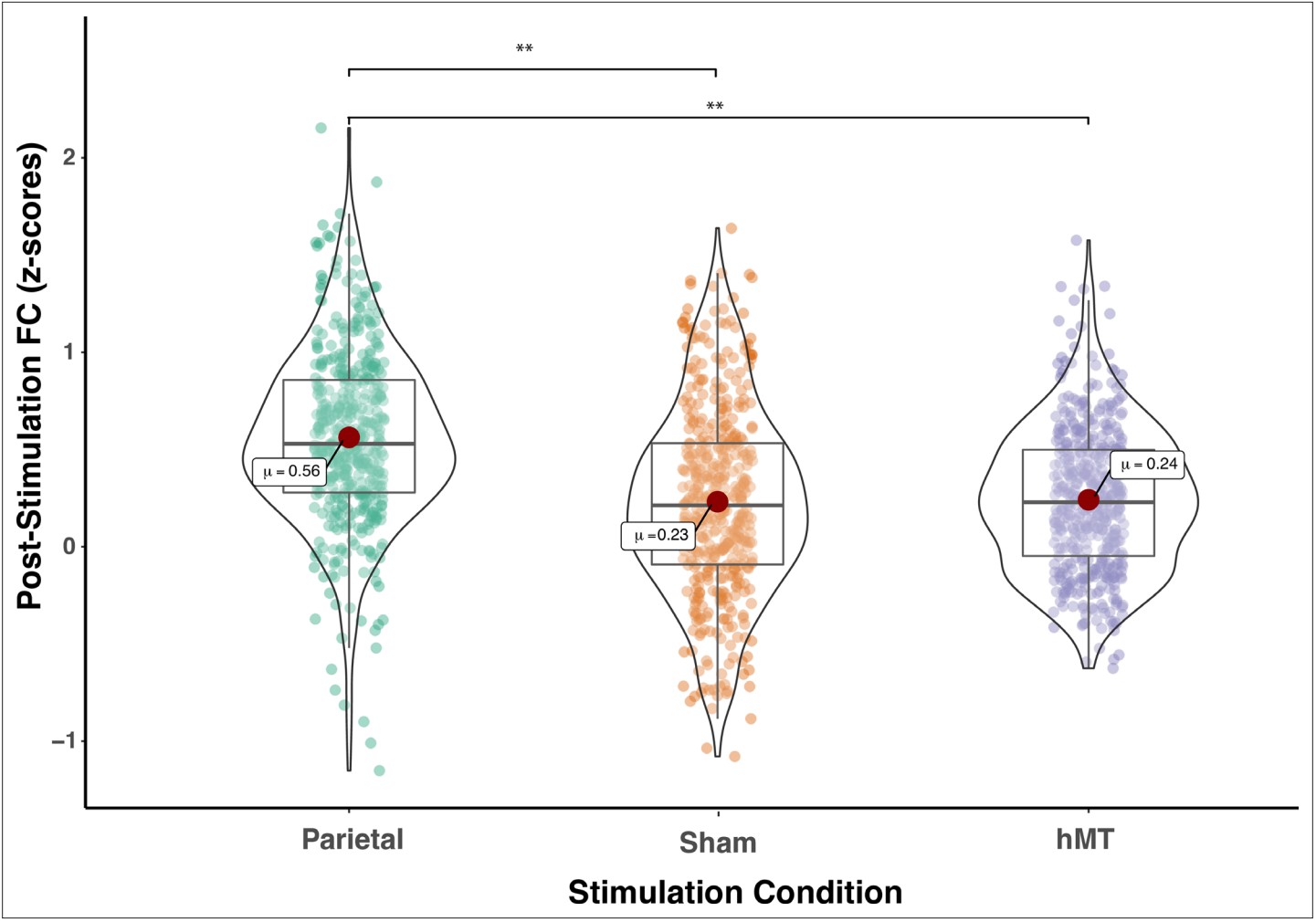

**Figure 5.** Functional connectivity within the DVAN after multi-session hf-tRNS coupled with training (post-test session). Correlations of all ROI pairs (z-scores) constituting the DVAN are represented separately for the three stimulation conditions (Parietal, Sham, and hMT+). Asterisks represents significant differences (**p<0.01). DVAN, dorsal and ventral attention network; ROI, region of interest; tRNS, transcranial random noise stimulation.

homogeneity of variances was violated, the Brown-Forsythe F-ratio is reported. This analysis revealed a strong significant effect of stimulation condition on post-test FC scores (F(2, 1286.64)=81.839, p≤0.00; $\eta_p^2$=0.108), which indicates that FC was significantly different on this session among the three stimulation conditions (*Figure 5*). T-test comparisons on post-test FC scores revealed that connectivity was significantly increased in the Parietal group compared to the hMT+ (t(865.23) = 11.59, *P* =< .001, df corrected for inhomogeneity of variance; Cohen's d = .773) and to the Sham group (t(892.6)=10.518, p<0.001, df corrected for inhomogeneity of variance; Cohen's d=0.701). Post-test FC scores were not significantly different between the hMT+ and the Sham groups (t(837.68)=0.357, p=0.721, df corrected for inhomogeneity of variance; Cohen's d=0.024).

In addition, we further investigated the modulation of the DVAN after stimulation and we computed the difference between FC scores pre- and post-test sessions (ΔFC=FC(S2)−FC(S1); *Figure 4*, right panel) to highlight the impact of stimulation on all ROIs-to-ROIs connections within the network. We run t-test comparisons on the delta FC scores between condition and we found scores were significantly different in the Parietal group compared to the hMT+ (t(890.801)=11.443, p≤0.001; Cohen's d=0.763) and to the Sham group (t(879.469)=10.707, p≤0.001; Cohen's d=0.714), while FC did not differ significantly between Sham and hMT+ groups (t(852.124)=0.594, p=0.552; Cohen's d=0.040).

*Figure 4* shows how connections between the nodes of the DVAN changed after Parietal stimulation resulting in stronger positive correlations within the network nodes, while the connections

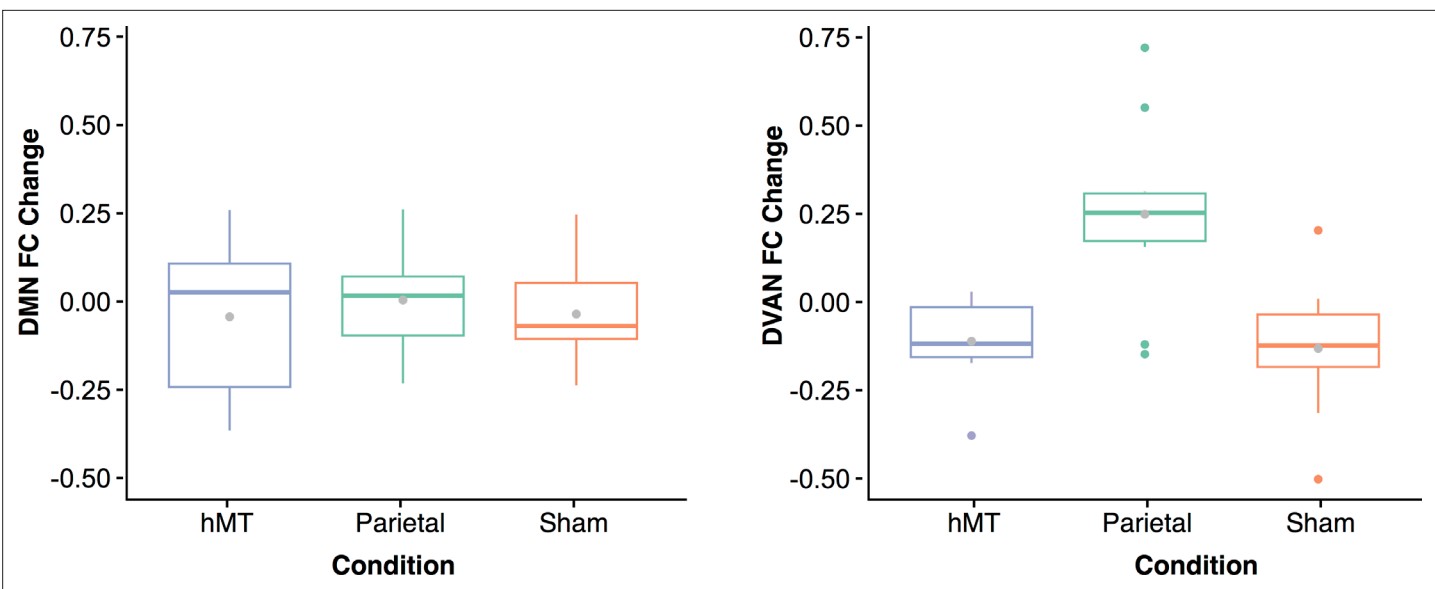

**Figure 6.** Functional connectivity (FC) changes between pre- and post-stimulation. Box plots for overall change in functional connectivity (calculated as the difference in connectivity between the post- and the pre-stimulation session) for the default mode network (**A**) and the dorsal ventral attention network (**B**) depicted per stimulation condition (Parietal, hMT+, and Sham). Gray dots within each box represent the mean change in FC per condition.

between the nodes after hMT+ and Sham stimulation resulted in a more negative correlation, indicating decreased connectivity .

Finally, to control for network-specific effects of stimulation, we also computed changes in rs-FC within the DMN which we hypothesized not to be affected by our stimulation protocol. This control analysis found no significant main effect of stimulation condition, nor interaction between stimulation condition and training. Specifically, in the first type of connectivity analysis when the network was taken as a whole (averaged connectivity representing the network), we found no effect of stimulation by site ($F_{(2, 27)}=0.09$, $p=0.91$), nor by session ($F_{(1, 27)}=0.58$, $p=0.45$), and no interaction between stimulation condition and session ($F_{(2, 27)}=0.20$, $p=0.81$). We then compared the mean connectivity modulation measured within the DMN with the mean modulation measured within the DVAN, that is the difference in connectivity between the pre- and post-stimulation session calculated for each network (*Figure 6*). A two-way ANOVA was conducted to examine the effects of network and condition on the mean connectivity modulation measured within the DMN and the DVAN. This analysis revealed a statistically significant interaction between the effects of network and stimulation condition factors on FC modulation scores ($F_{(2, 54)}=4.97$, $p=0.010$, $\eta_p^2=0.156$), as well as a significant effect of the stimulation condition factor alone ($F_{(2, 54)}=7.937$, $p\leq0.001$; $\eta_p^2=0.227$). The network factor alone was found to be not significant ($F_{(1, 54)}=0.274$, $p=0.603$, $\eta_p^2=0.05$). Simple main effects analysis showed that FC was modulated differently between conditions within the DVAN ($p<0.001$) but not within the DMN ($p=0.836$), and that FC modulation was different for the Parietal condition only ($p=0.006$), while it did not differ significantly for the Sham ($p=0.256$) nor the hMT+ conditions ($p=0.418$).

We also conducted a node-by-node FC analysis (all ROIs pairs within the network as single components of the network) and found no effect of session ($F_{(1, 1347)}=2.4$, $p=0.116$), nor of stimulation site ($F_{(2, 1347)}=0.347$, $p=0.707$). The interaction between stimulation group and session was also not significant ($F_{(2, 1347)}=0.853$, $p=0.426$; see Appendix for further details and additional analysis). *Figure 7* (left panel) shows how connections did not change within the DMN nodes as the mean of the z-score distribution for the pre- and post-test session is similar for all three groups and almost overlaps for the Parietal group. This is also evident from the correlation matrices (*Figure 7*, right panel) that represent the difference between FC scores pre- and post-test sessions for all ROIs pairs included in the network. The lack of connectivity modulation within the DMN and the specificity rs-FC modulation found after parietal stimulation only, provide strong evidence for a selective stimulation-induced effect by our stimulation and training protocol.

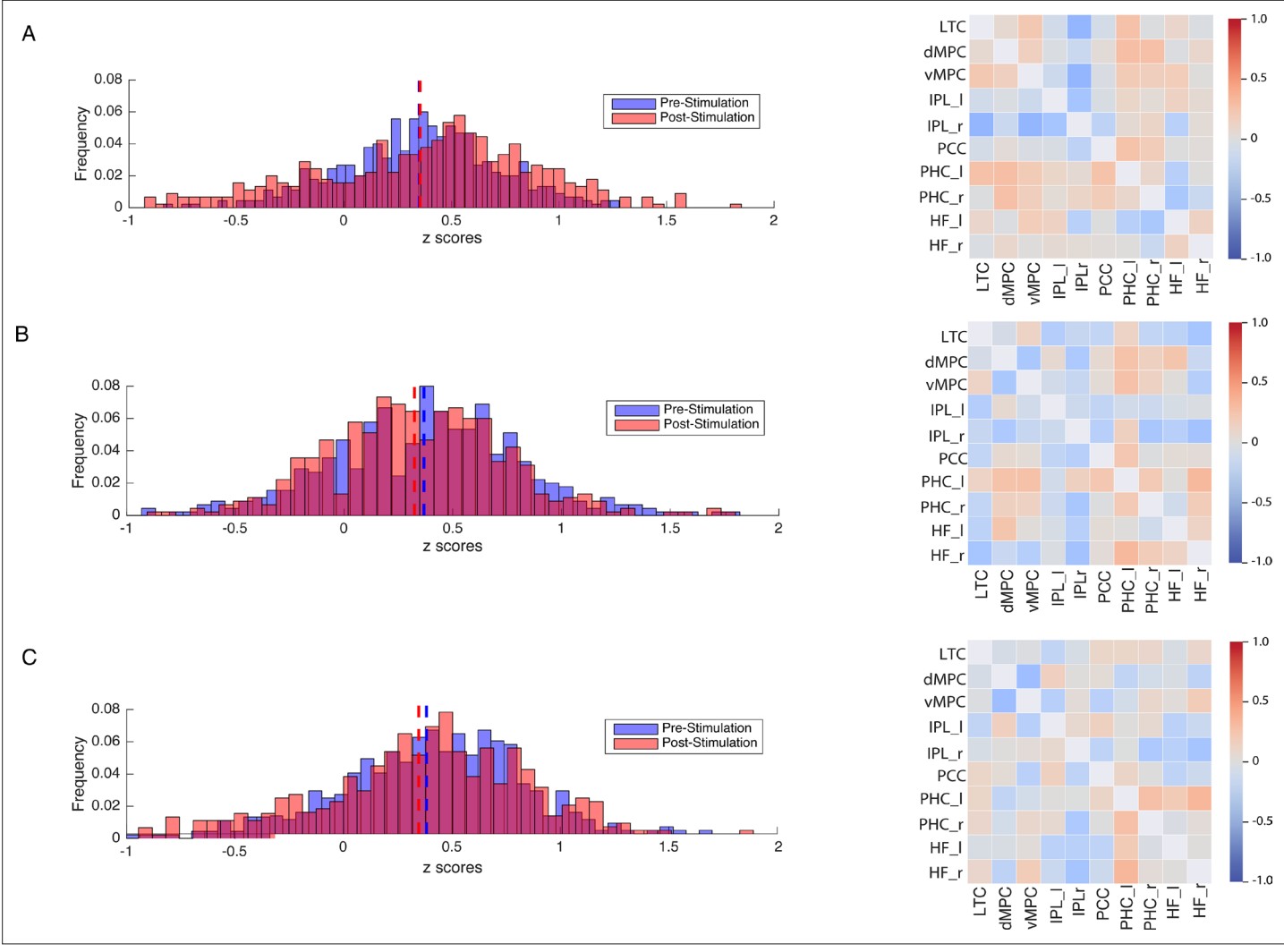

**Figure 7.** Modulation of resting-state functional connectivity of the DMN. Left Panel: z-scores frequency distribution pre- (light blue bars) and post- (light red bars) multi-session hf-tRNS coupled with training per each stimulation condition (A. Parietal, B. hMT+, C. Sham). Dark red indicates the overlapping distribution between pre- and post-sessions. Dotted red and blue lines indicated the mean for each z-scores distribution. Right Panel: Correlation matrices represent the result of the computed difference between FC scores pre- and post-test sessions (Δ=FC(S2)−FC(S1)) depicted by condition (A. Parietal, B. hMT+, C. Sham). Each correlation difference between ROI pairs was first calculated at subject level and then averaged across subjects. Colors bars indicate the strength and direction of correlation change values for each regions' pair (red colors indicate higher connectivity; blue colors indicate lower connectivity). DMN, default mode network; ROI, region of interest; tRNS, transcranial random noise stimulation.

## Functional connectivity-behavior correlation

Next, we examined the relationship between changes in behavioral performance and the resting-state FC within the DVAN. Only those subjects that were included in both behavioral and FC analysis were incorporated in the linear model analysis (a total of 26 subjects). We observed a number of positive correlations between FC changes and behavioral improvement in the OD task. In particular, to estimate the behavioral relevance of rs-FC, we correlated connectivity scores with two independent behavioral indices: one behavioral index measured during the scanning sessions (pre- and post-test sessions), and a second behavioral index measured during the Training sessions. We found subjects that displayed a higher performance improvement also tended to show higher rs-FC within the DVAN (measured as the mean of the correlations within the nodes of the DVAN) in the post-test session (r=0.44). The linear regression model revealed a significant regression coefficient (p=0.022, R-squared=0.2) which indicates that post-session functional connectivity scores are significantly associated with changes in behavioral scores in the final test session. We then examined whether there was a relationship between rs-FC and behavioral changes measured at the end of training (Training

Day 1 relative to Training Day 4). We found that subjects with higher FC scores within the DVAN in the post-test MRI session showed a positive correlation with higher behavior improvement in the OD task achieved during training (r=0.48). The regression coefficient for this relationship was significant (p=0.012, R-squared=0.23) indicating that changes in behavioral performance during training are significantly correlated with post-test session connectivity scores. Taken together, these results indicate that post-test functional connectivity scores within the attention network significantly correlated with behavioral improvement measured not only immediately at the end of the stimulation and training protocol, but also 24 hr after the end of the stimulation protocol.

## Discussion

In this study, we investigated the potential for visual training coupled with noninvasive brain stimulation to promote significant improvements in behavior and increase cortical functional connectivity. We trained subjects on cross-tasks training to optimize learning in an attention-based learning paradigm over multiple days. We combined this training with high frequency tRNS using a protocol optimized to facilitate cortical plasticity with the goal of increasing and speeding up training improvements relative to training without tRNS. Our stimulation protocol, when applied over parietal cortex, dramatically shortened the learning period, overcoming the limitations of previous training paradigms. Crucially, tRNS coupled with training also increased functional connectivity within the attention cortical network when delivered over parietal cortex, and the connectivity pattern changed more strongly for individuals with the largest behavioral improvement during and after training. Thus, we conclude that tRNS has the potential to strengthen task-relevant connections between sensory and attention systems to promote learning and plasticity.

Bilateral tRNS over the parietal cortex during training resulted in a strong boost in performance, while tRNS over hMT+ and Sham had no effect on behavior. Crucially, improvement in behavior in the parietal condition positively correlated with the increase in functional connectivity within the main nodes of the DVANs, indicating that concurrent training and stimulation modulated large-scale cortical dynamics, strengthening functionally selective neural pathways (*Chiappini et al., 2018*), a potential physiological marker of enduring changes (*Polanía et al., 2011*).

We also sought to understand whether a significant improvement could be achieved with a short training protocol, in conditions where we normally would not expect learning to occur (*Huang et al., 2012*; *Aberg et al., 2009*; *Li et al., 2004*). We adopted a cross-task training that uses the same visual stimulus for two different tasks (*Szpiro et al., 2014*), and we coupled the training with a stimulation protocol during the learning acquisition phase. While during the Sham and hMT+ control conditions no learning was achieved, our data suggest that it might be the combination of a challenging task with concurrent brain stimulation over the relevant parietal cortical circuits that facilitated fast learning, a paradigm amenable to promote plasticity (*Herpich et al., 2019*; *Tyler et al., 2018*; *Cappelletti et al., 2013*; *Kiyonaga et al., 2021*).

We next examined the time course of the stimulation effect upon behavior across the four short training sessions. tRNS administered on the IPS during training promoted fast learning, with performance in the parietal group significantly diverging from the control groups by the third day of training. Notably, the Parietal tRNS group displayed a build-up and cumulative trend of the facilitation effect, as accuracy increased on each training day and retained the beneficial effect acquired in the previous session. This facilitation effect was not only quantitatively different from the other groups, but also selective for one task (the OD visuospatial task), within the same group of participants. These data confirm the previously observed beneficial effect of tRNS over training (*Cappelletti et al., 2013*; *Herpich et al., 2019*). However, they further indicate that tRNS's enhancing properties are mostly effective when stimulation is coupled with a relevant stimulus that maximizes the related cortical response, like in our OD task (*Kastner and Ungerleider, 2000*).

Taken together, these behavioral results suggest that learning of a visuospatial attention task was achieved within a few training sessions and that this facilitation was long lasting, as increased performance outlasted the end of stimulation for at least 24 hr, when the last fMRI session after the end of training was performed. These findings extend the results of a previous experiment that tested tRNS efficacy upon attention within one session, but found a rapidly decaying beneficial effect by the end of stimulation (*Tyler et al., 2018*). However, using a multi-session stimulation approach coupled

with training promoted retention of the beneficial effect of training, as also shown by the results in increased rs-FC after training (*Chen et al., 2015*; *Sarabi et al., 2018*; *Kang et al., 2018*).

There might be several conditions that facilitate the emergence of learning, all likely driving the stimulus-related activity to reach the learning threshold (*Seitz and Dinse, 2007*). First, in this study, we demonstrate that stimulation site selection within a network is crucial to obtaining successful augmentation. Performance improved only when tRNS was delivered to the parietal cortex, and was absent in all other conditions, including stimulation to hMT+, often included as a node of the dorsal attention network. Thus, our results indicate the parietal lobes have a direct involvement in visual PL and attention processes (*Giovannelli et al., 2010*; *Law and Gold, 2008*; *Shadlen and Newsome, 2001*; *Colby and Goldberg, 1999*). Second, these results indicate that tRNS effects do not spread to regions in close anatomical proximity, but rather spread within strongly functionally connected circuits. Stimulation to hMT+, which is close to the parietal lobes, did not have any effects on behavior nor on FC. Third, our findings support the idea that learning is facilitated when the cortical processes underlying the task are stimulated while in an aroused state derived by task performance (*Wright et al., 2010*), and indicate the possibility to use tRNS to rapidly reach this optimal state. Importantly, our study further indicates the benefits of short but effective training sessions, an ideal condition to promote learning (*Censor and Sagi, 2008*; *Censor et al., 2016*).

One might ask why we observed learning occurring for one attention task only. Our results are consistent with previous studies that used a similar cross-task training paradigm and found behavioral improvement in only one of two tasks trained (*Szpiro et al., 2014*; *Wright et al., 2010*). One potential explanation for task selective improvement is that making two different perceptual judgments on the same visual stimulus may require tuning of different cortical representations (*Li et al., 2004*). Simultaneous training might induce competition between these two representations, ultimately leading to the enhancement of one and inhibition of the other, as suggested in the augment Hebbian reweighting model (AHRW; *Dosher et al., 2013*; *Dosher and Lu, 2017*). Similar to Szpiro et al.' study, when our participants trained on two randomly interleaved tasks using identical visual stimuli, improvements were observed only in the OD, suggesting the training may have changed the perceptual weighting of the orientation channels.

In line with this hypothesis, our study showed that orientation PL was apparent with our very short training sessions only when coupled with tRNS over parietal cortex, further emphasizing the role of neuromodulation in speeding up the learning process. In the parietal stimulation group, we positioned the electrodes over the EEG locations P3 and P4 (*Herwig et al., 2003*; *Okamoto et al., 2004*), roughly corresponding to the posterior IPS. The IPS has been linked to visuospatial attention in several imaging and brain stimulation studies (*Corbetta and Shulman, 2011*; *Battelli et al., 2009*; *Battelli et al., 2017*), and the OD task involves the processing of visuospatial components. Spatial processing was not relevant in the TOJ condition and that is typically associated to more ventral areas (*Chica et al., 2011*; *Agosta et al., 2017*; *Husain and Nachev, 2007*). The temporo parietal junction (TPJ), at the crossroad between the ventral and the dorsal visual pathways, has been linked to temporal attention (*Tyler et al., 2015*: *Battelli et al., 2007*; *Husain and Rorden, 2003*). Although this explanation is rather speculative, the selectivity of the behavioral effect along with functional connectivity results might indicate that the IPS plays a pivotal role in OD across the visual space (*Capotosto et al., 2013*; *Corbetta and Shulman, 2002*). Moreover, the parietal lobe has been associated with selective and gating attention mechanisms (*Suzuki et al., 2013*), and tRNS might have facilitated these gating mechanisms during learning. Interestingly, the analysis of the current distribution during stimulation shows that the effects driven by parietal stimulation are likely due to an effective targeting of selective task-active crucial nodes (*Figure 8A and B*; ). With a similar pattern of distributed activity, OD and TOJ functionally activated the same task-based cortical regions at baseline (*Figure 8A*, details for statistical analysis are reported in the 'Task-evoked brain activity' section). However, the behavioral gain was dramatically different between the two tasks. This clearly shows that stimulus-evoked functional activity occurring at individual brain regions is not informative about what regions are crucial for learning. As shown in previous studies, regions part of the same network may flexibly interact to perform and prioritize different tasks (*Gratton et al., 2016*). In our data, functional connectivity changes among cortical areas showed unique pattern of spontaneous response at rest, after parietal stimulation only, and tRNS might have enhanced these critical changes in connectivity among distributed brain regions during learning (*Cole et al., 2013*).

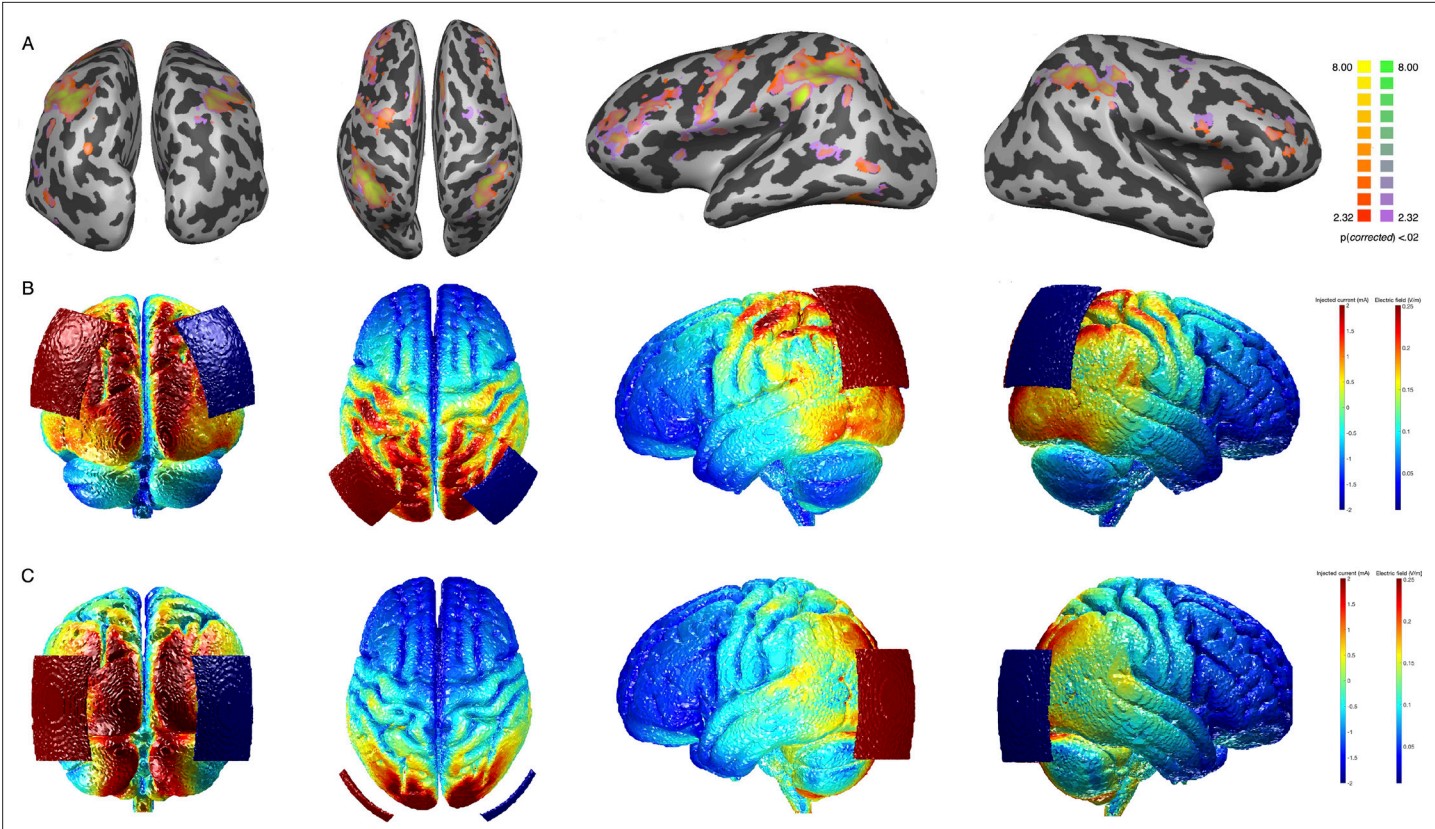

**Figure 8.** Task-related activity for the orientation discrimination (OD) task and electric field distribution after stimulation. Panel A: group GLM contrasting neural activity in OD and TOJ trials versus rest (see Task-evoked brain activity section). All regions are significant at q<0.05 (FDR corrected). Yellow and red (for the OD task) and yellow and purple (for the TOJ task) colors indicate significantly positive activation. Panel B: electric field distribution on the brain induced by Parietal stimulation. Panel C: electric field distribution on the brain induced by hMT stimulation. FDR, false discovery rate; GLM, general linear model; TOJ, temporal order judgment.

Finally, some studies have analyzed the importance of task engagement or difficulty during stimulation, and have demonstrated how these factors affect the behavioral outcome (*Bortoletto et al., 2015*; *Hsu et al., 2016*). In our study, participant engagement was challenged in two ways. First, task difficulty was individually tailored and equated for difficulty across subjects ensuring subjects' engagement with the task (*Waskom et al., 2019*). Second, the experimental design randomly alternated between experimental tasks, forcing the continuous switch of attention between stimulus features, might force a strategy that favors one task due to limited available cognitive resources.

We next asked whether there were cortical physiological changes that indicate more sustained plastic changes along with behavioral change, and if such changes were within task-related well-established attention-related cortical networks. Specifically, we analyzed whether the multi-session tRNS had an effect on functional connectivity at rest in the DVAN, which we targeted through stimulation. Brain regions in this network mediate attention mechanisms and consequently shape the analysis of visual perceptual input (*Kastner and Ungerleider, 2000*). The dorsal/ventral attention network plays a pivotal role in attention functions and constitutes a large occipital-frontoparietal network that controls allocation of attention to positions and visual features in space (*Kastner and Ungerleider, 2000*; for a review see *Fiebelkorn and Kastner, 2020*: *Vossel et al., 2014*). Tasks requiring selective attention have been shown to activate the ventral part of the attention network (VAN; *Corbetta et al., 2008*; *Vossel et al., 2012*), while the dorsal portion of the attention network, in particular the IPS and the FEF regions, is active during many visuospatial tasks that require spatial updating and the selection of important regions of space (*Kastner and Ungerleider, 2000*; *Corbetta et al., 1998*; *Merriam et al., 2003*; *Greenberg et al., 2010*).

Consistent with the relevance of both ventral and dorsal streams of the attention network for the performance of the task used in this experiment, our results show increased functional connectivity within the entire network. Almost all connections between the node pairs displayed higher connectivity after multi-session parietal stimulation only. This result indicates that bilateral hf-tRNS over the parietal lobes modulated the whole attention-related network, instead of producing local changes alone. Importantly, the neural modulation was network-specific, not a propagation of stimulation across neighboring cortical regions or networks. The control analysis we conducted on the DMN showed no changes in connectivity following stimulation and training, indicating that our stimulation-induced changes were selective for the DVAN. Moreover, the node-selective effect within the DVAN was demonstrated in the hMT+ stimulation group, where the neighboring IPS was not modulated by hMT+ stimulation. Similar to our results, previous physiological studies on non-human primates found changes in the lateral intraparietal area but not in the MT area suggesting that PL requires the involvement of higher areas that induce changes in how the sensory information is interpreted to drive behavior (*Law and Gold, 2008*). The lack of improvement in the hMT+ stimulation condition might indicate that while this cortical area might be involved in attentional processing (*Yates et al., 2017*; *Yao et al., 2016*), for other functions involved in learning, such as decision processing, the parietal lobe plays a more pivotal role (*Herrington and Assad, 2010*; *Zhou and Freedman, 2019*). Our results are also consistent with previous research that found persistent network dynamics changes following other types of stimulation (e.g., transcranial magnetic stimulation, TMS; *Battelli et al., 2017*; *Ruff et al., 2006*; *Ruff et al., 2008*; *Lee and D'Esposito, 2012*), and points to the possibility to use our neuromodulation protocol to exert long lasting and distal effects. Finally, one potential limitation of our study is the relatively small sample size, given all the conditions we tested. As previously pointed out, brain-behavioral correlations can produce inflated effect sizes with the combination of relatively small sample size and stringent alpha-correlations (*Yarkoni, 2009*).

Functional connectivity changes following parietal stimulation are likely to have affected behavior and driven the system to better respond to the demands of the attention task. Rapidly evolving changes in connectivity between interconnected regions have been proposed as the key mechanism that allows large-scale networks to respond to task demands (*Buschman and Kastner, 2015*). In the context of tRNS, we speculate that a fast modulation of the network connectivity can be due to the temporal summation of neural activity, as well as to the offline stochastic effect of tRNS (*Chaieb et al., 2015*). In fact, the stochastic resonance mechanism has been proposed to lead to higher synchrony of oscillations between neurons across large-scale functional systems, which in turn creates strong links between firing neurons (*van der Groen and Wenderoth, 2016*; *van der Groen et al., 2018*; *Schwarzkopf et al., 2011*; *McDonnell et al., 2009*; *McDonnell and Ward, 2011*). The strong correlation found between improved performance and increased FC suggests the pivotal role of the IPS in network mediating these system effects during PL (*Szczepanski and Kastner, 2013*).

## Conclusions

This multi-method study investigates the potential long-term benefits of tRNS on cortical plasticity and its ability to efficiently alter cortical networks to promote attention and visual PL. Our neuroimaging and behavioral results support the idea that the modulation of task-related neural networks induced by tRNS can efficiently promote behavioral improvements and speed up the emergence of learning. Overall, this is the first study demonstrating the sustained and selective nature by which tRNS operates on cortical dynamics, and consequently opens a critical window during which the cortex might be more plastic and responsive to reorganization needs to promote learning. Clinical and healthy populations could significantly benefit from the shortened learning time afforded by the combination of tRNS with behavioral learning.

## Materials and methods
### Regulatory approval
The study was approved by the ethical committee of the University of Trento.

## Participants

Thirty-seven neurologically healthy subjects (20 females, mean age 22.8 years old), with normal or corrected-to-normal vision participated in the study. All subjects were screened for medical contra-indications to MRI and brain stimulation and received monetary compensation for their participation in the experiment.

## Study design

Participants were randomly assigned to one of three stimulation conditions: (1) bilateral tRNS over parietal cortex (Parietal group), (2) bilateral tRNS over middle temporal cortex, hMT+ area (hMT+ group), and (3) Sham stimulation (Sham group). Seven subjects were excluded from the analysis due to head motion during one or both scanning procedures or due to inadequate behavioral performance (i.e., performance was at chance in catch trials), leaving 30 participants for behavioral and FC analysis.

Subjects participated in a multi-session experiment that lasted 7 days (one session per day; *Figure 1A*). On thresholding session, they completed a staircase procedure to measure their psycho-physical thresholds for two separate attention tasks: a TOJ and an OD task (described below). On pre-test session, resting-state and the attention task performance measures were taken in the MRI, with trials (from both tasks) randomly interleaved within each block. Participants completed a total of five blocks per task (see Neuroimaging procedure section for details). On training and stimulation sessions (Training Day 1, Training Day 2, Training Day 3, and Training Day 4), subjects received 25 min hf-tRNS or sham while training on the same TOJ and OD tasks, with the two trial types randomly interleaved (cross-tasks training), as in the pre-test (five blocks per day). Post-test session was a repeat of pre-test session.

## Stimuli

Stimuli consisted of a pair of sine-wave gratings (Gabors) positioned 4° to the left and right of a central fixation cross. The Gabors were presented with a spatial frequency of 3 cycles/deg and reduced 50% contrast against a uniform gray background (47.5 cd/m²), with a fixation cross positioned in the center of the screen for the entire duration of the trial. The two Gabors were presented with temporal offset asynchronies (for the TOJ task) that corresponded to the individual temporal threshold described in detail below. In addition to the temporal offset, one of the two discs was tilted along the vertical line with a degree that corresponded to the individual orientation threshold calculated for the OD task at baseline. In the TOJ task, participants were asked to judge whether two Gabors were presented simultaneously or not, while in the OD task, participants were asked to determine whether the two Gabors had the same or different orientation. Throughout the experiment, subjects always viewed stimuli at their individual thresholds (except during catch trials where stimuli were presented above threshold) for both dimensions (time and orientation); however, on any given trial, they had to attend to only one of the features, as instructed by the pre-cue word. The target feature (or 'test Gabor') was presented for equal number of trials (120 trials per trial type, 240 in total per session) on the left and right visual field, within each block per task. Easy (above threshold) catch-trials were also interleaved randomly within each block (n=6, 24 catch trials per session) to check that participants remained engaged and were performing the cross-attention task correctly. In the Easy trials, the two Gabors were presented with large offset asynchronies (≈150 ms, for the TOJ condition), or with a large orientation difference (≈20°, for the OD condition). Importantly, in both tasks, visual stimuli were presented with temporal offsets (TOJ) and orientation values (OD) fixed at individual threshold levels, which were calculated on thresholding session.

During training days, stimuli were displayed on a 22-in. LCD monitor with a 60 Hz refresh rate controlled by a DELL computer equipped with Matlab r2016a (The MathWorks, Natick, MA) and Psychtoolbox 3.0.8 (*Brainard, 1997*; *Pelli, 1997*). Participants were seated 57 cm from the screen in a dark and quiet room, and used a chin rest to ensure consistent positioning. During the fMRI sessions (pre- and post-test sessions), stimuli were displayed on a Nordic NeuroLab LCD monitor (Basic monitor specs include: 878 mm horizontal × 485 mm vertical; 3840×2160 pixels) connected to a Windows PC equipped with the Matlab r2016a and Psychtoolbox 3.0.8, and stimuli were back projected through a mirror inside the scanner.

## Apparatus and procedure

On thresholding session (*Figure 1A*, green section), subjects performed two two-alternative forced-choice tasks within each block: (1) a TOJ or (2) an OD task. Two 3–1 staircase procedures were used to assess thresholds values for the two tasks (TOJ and OD), separately. A single incorrect response decreased task difficulty, while three consecutive correct responses increased task difficulty. The staircase terminated after 30 reversals of the staircase parameter. Temporal offsets and orientation values in the remaining sessions were calibrated based on the thresholds to yield 50% accuracy in the same/different judgments. The threshold corresponded to the point of subjective equality, which represents the threshold value at which the observer experiences two stimuli as identical. The estimated threshold was then used for the last 15 trials and it was considered accurate if subjects performed at 50% on these trials, the expected performance. 50% accuracy was chosen to prevent ceiling effects, thus allowing PL to occur.

On the pre-test, post-test, and training sessions, participants performed the PL cross-tasks procedure. Each trial started with a 2-s instruction interval during which a cue-word indicated which of the two tasks the subjects should perform in the upcoming trial: 'Time' instructed the subject to perform the TOJ task (120 trials), and 'Orientation' instructed the subject to perform the OD task (120 trials). After the pre-cue, stimuli were presented for 500 ms. Subjects were then prompted to make a forced-choice judgment to indicate whether the two stimuli had same/different orientation (OD task) or if they were presented at the same/different time (TOJ task), depending on the task. During the response interval (1.5 s), a cue-word was presented to remind subjects which judgment they had to report. A fixation interval of 2 or 4 s terminated the trial (trial example, *Figure 1D*).

## Stimulation protocol

Before testing, all participants were provided with a short introduction to brain stimulation and safety information. After each participant was briefed, they completed a stimulation safety questionnaire and signed the informed consent. At this time, any participant deemed ineligible for stimulation or MRI procedures was excluded from the experiment. A battery-driven stimulator (DC-Stimulator, Neuro-Conn, Ilmenau, Germany) was used for electrical stimulation. For the two active tRNS conditions (Parietal and hMT+), 2 mA current was applied for 25 consecutive min with random alternating frequency delivered at a high frequency range between 101 and 640 Hz. Stimulation was delivered with a fade in/out period of 20 s at the beginning and at the end of each stimulation session. For Sham stimulation, the machine was turned off after the fade-in phase. Two rubber electrodes (size=5×7 cm$^2$), contained in sponges soaked in saline solution, were placed on the subject's head and were kept fixed on the stimulation sites with a rubber band and a head cap. Stimulation sites were identified using the International electroencephalographic 10/20 system for scalp electrode localization. The center of the electrode was placed bilaterally over PO7/PO8 (left and right, respectively) for hMT+ and bilaterally over P3/P4 for parietal and sham conditions (*Figure 1B*). Participants reported no noticeable sensation resulting from tRNS (see also *Ambrus et al., 2010*). Modeling of the electric field following tRNS was performed with free ROAST software package (*Huang et al., 2019*; *Huang et al., 2016*) using the standard MNI-152 template (ICBM project) and visualized separately for the two active stimulation groups (*Figure 8B and C*). To evaluate whether the electric fields peaks induced by the two stimulation conditions targeted the same brain regions activated by the behavioral tasks, we analyzed BOLD signal changes in response to the OD and the TOJ task in the pre-stimulation session (details for statistical analysis are reported in the Task-Evoked brain activity section). *Figure 8* includes the results of a second-level deconvolution general linear model (GLM) analysis for significant positive BOLD response for the OD and TOJ tasks versus rest (*Figure 8A*), together with the simulation of the current distribution induced by the two stimulation conditions to qualitatively assess the overlap between positive BOLD response for the two tasks and the current distribution peaks over the cortex.

## Neuroimaging procedure

Whole-brain scanning was performed with a 4 T Bruker MedSpec MRI scanner using an 8-channel head-coil at the Center for Mind and Brain Sciences of the University of Trento, Italy. High-resolution T1-weighted images were acquired for each subject at the beginning of each fMRI session (MP-RAGE 1×1×1 mm$^3$ voxel size, 176 sagittal slices). Functional images (T2*-weighted EPIs, TR=2.0 s, TE=28.0, flip angle=73∘, 3×3×3 mm$^3$ voxel size, 0.99 mm gap, 30 axial slices acquired interleaved, 192 mm

FOV) were collected, for a total of 120 volumes per resting-state functional run and 185 volumes per task-based functional run.

The entire session consisted of one anatomical run followed by six functional runs (one resting-state run, five task runs collected on the same day), using the following procedure: first, a high-resolution anatomical image was acquired, followed by three functional task runs, one resting-state functional run, and two additional functional runs (same order on each session). The resting-state functional run was set in-between the task-based functional runs to allow participants to rest during the fMRI session that overall lasted about 1 hr.

During MRI data collection, subjects viewed the stimuli through a periscope mirror mounted on the MR head-coil that allowed the subject to view a screen positioned at the head of the scanner. Throughout the experiment, participants were instructed to maintain fixation on a constantly present central fixation cross. Participants were instructed to indicate the response with an MR compatible response box (two double-buttons response pads) with left index finger indicating 'same' and right index finger indicating 'different' responses.

## Data analysis

### Behavioral data analysis

Statistical analyses were performed using MATLAB (The MathWorks, Natick, MA) and SPSS (IBM Corporation, Armonk, NY), while figures were made using both MATLAB and R toolbox (ggplot2, CRAN; *Patil, 2018*). We first compared behavioral data from pre- and post-test days (see 'Magnitude of learning' section), the two sessions during which fMRI data were also collected, while data from the training and stimulation phase (Training Day 1, Training Day 2, Training Day 3, and Training Day 4) are presented separately (see 'Time course of learning' section). Accuracy scores were calculated as a proportion of correct responses from the recorded responses on each day. Data are analyzed and depicted for each task (OD and TOJ) and each analysis (pre- vs. post-test analysis, and training analysis) separately.

For pre- and post-test data analysis, we first checked for normality and homogeneity using Levene's test and corrected with Brown-Forsythe correction when necessary. We performed two mixed repeated measures ANOVA to determine the effect of stimulation condition on performance in the two attention tasks across sessions. The within-subjects factor was Day ($n = 2$ for pre-post stimulation analysis) and the between-subjects factor was Stimulation condition ($n = 3$). For the pre- and post-test analysis, we tested whether the three stimulation groups differed at pre-test (baseline) and post-test sessions (post-stimulation and training session). Pairwise comparisons were performed to compare pre- and post-test performance for each group. We further analyzed data by calculating the total improvement as the difference between the post- and pre-test performance, thus obtaining the delta value for each stimulation condition (ΔPerformance|Improvement=Post-Test Performance − Pre-Test Performance). We then performed a one-way ANOVA to test whether there was a significant difference between the performance improvements (delta values) between stimulation conditions, and run post hoc pairwise comparisons to reveal differences in improvement between groups. Next, we analyzed data from the training and stimulation phase for the two tasks separately. Specifically, to measure changes in performance over time, we performed a mixed repeated measures ANOVA with stimulation condition ($n = 3$) as between-subjects factor and Training Day (Time, $n = 4$) as within-subjects factor. Mauchly's test was used to evaluate the assumption of sphericity, and, when necessary, the Greenhouse-Geisser correction was used to correct degrees of freedom and to assess the significance of the corresponding p-values, while Levene's test was used to address the assumption of equality of variances. Following the significance found in the mixed repeated measures ANOVA, we then compared performance between stimulation groups on each training day, and run post hoc pairwise comparisons to reveal simple differences between stimulation groups and training days performances.

### Resting-state fMRI data analysis

We analyzed resting-state data collected in each fMRI session (pre-post sessions). Pre-processing of functional scans was initially performed using BrainVoyagerQX (Brain Innovations Inc, Maastricht, The Netherlands) and included the elimination of the first four volumes due to non-steady-state magnetization. Images were subsequently subjected to 3D motion correction, slice-timing correction, realignment, and spatial smoothing with a 6-mm FWHM Gaussian kernel. Runs with instantaneous head

**Table 1.** Brain coordinates.

Group mean Talairach X, Y, and Z coordinates for the centroid of each region of interest (left and right hemisphere) for the DVAN.

| | | ROIs standardized coordinates | | | | | |
| --- | --- | --- | --- | --- | --- | --- | --- |
| | | Left Hemisphere | | | Right Hemisphere | | |
| | | X | Y | Z | X | Y | Z |
| DVAN | IPS | −26 | −63 | 48 | 25 | −60 | 48 |
| | hMT+ | −44 | −67 | 3 | 44 | −71 | 6 |
| | FEF | −30 | −5 | 48 | 30 | −5 | 48 |
| | TPJ | −48 | −54 | 26 | 48 | −54 | 21 |
| | VFC | −42 | 12 | −1 | 40 | 17 | 4 |

motion exceeding 3 mm in any given dimension were discarded. The functional scans were manually co-registered to each individual's high-resolution anatomical scan, then normalized into standardized Talairach coordinates (*Laitinen, 1989*).

Advanced pre-processing of the functional data from targeted ROIs was performed in MATLAB (The Math Works) to remove data contamination caused by motion, which can alter functional connectivity scores. This pipeline included nuisance regression using 12 motion estimate regressors (the six head realignment parameters obtained by rigid body head motion correction and their first-order derivatives) and estimates of global signal obtained from the ventricles. The ROI time series were then detrended and volumes with movement-related activity were removed (scrubbing) and temporarily replaced with resampled volumes (computed using LSSD Fit) in order to apply band-pass filtering (0.009–0.08 Hz;). The censored and resampled volumes were then removed from further analysis. After these processing steps were applied, only those runs that retained at least 3 min worth of data were used for further analyses (*Satterthwaite et al., 2013*). Statistical analyses were performed to ensure that the number of censored frames was equivalent across conditions and sessions.

We computed functional connectivity on resting-state data using 10 bilateral ROIs included in the DVAN: the IPS, hMT+, and frontal eye fields (FEFs) of the dorsal network; and the temporal parietal junction (TPJ) and ventral frontal cortex (VFC) of the ventral network. These regions were identified using standardized mean coordinates (*Table 1*) derived from the literature (*Battelli et al., 2017*; *Vossel et al., 2014*; *Corbetta et al., 2008*; *Fox et al., 2006*). Functional data from all voxels within a 6-mm radius sphere from the center coordinate were averaged into a single time series, keeping the ROIs similar in size across participants. The time courses were extracted from each ROI in the resting-state run and functional connectivity scores were computed as the Pearson's r correlation coefficient. Correlation coefficients were Fisher-z transformed prior to subsequent statistical analyses.

The Pearson correlations (r scores) among all the ROIs embedded in the attention network produced a connectivity matrix for each subject and each session (pre-post session). In total, there were 45 unique connectivity values among the 10 ROIs (10*(10−1)/2) for each correlation matrix. Z-transformed correlations were used to compute several measures of functional connectivity.

In the first set of analysis, we analyzed whether stimulation-induced connectivity changes at a whole-network level by examining connectivity scores that were the result of averaged correlation values among all connections between ROIs (*van den Heuvel et al., 2017*; *Nicolini et al., 2020*). Hence, these analyses examined one connectivity score representing the overall connectivity trend of the entire network. To test the effect of training and stimulation site, functional connectivity scores were subjected to a mixed repeated measures ANOVA with the within-subject factor session ($n = 2$) and between-subject factor of stimulation site ($n = 3$). We also compared rs-FC at each time point (pre- and post-test sessions) by performing two one-way ANOVAs, one per session. Following significance found in the post-test session only, post hoc pairwise comparisons on post-stimulation connectivity scores were run to reveal simple effects between pairs of stimulation conditions. We then run the second set of analysis to examine whether all ROI-to-ROI patterns within the network reliably contribute to the overall condition-related differences in FC, and to evaluate whether there still was a significant change in connectivity modulation when taking into consideration the contribution given

by each pair of ROIs embedded in the network. We therefore adopted a different approach to analyze connectivity within the attention network where each ROI-to-ROI contribution (measures of connectivity level between two pair of ROIs as z-scores) to the network state connectivity is considered as a single observation of the attention network. We derived correlation coefficients within-subjects first, and then averaged across subjects obtaining mean z-scores per group, to ensure that changes in connectivity modulation were present in most individual subjects rather than derived from a group average (*Varangis et al., 2019*; *Bellana et al., 2017*; for further discussion about differences between within-subjects versus group-averaged approaches, see *Roberts et al., 2016*). In these analyses, we first performed a one-way ANOVA to compare connectivity scores between stimulation groups prior to stimulation (at baseline). We next tested the effect of stimulation coupled with training on functional connectivity by performing a mixed repeated measure ANOVA to test the effect of stimulation site and session on all correlation values within the network. We then compared post-test connectivity scores of all network components by performing a one-way ANOVA, followed by post hoc pairwise comparisons to reveal differences between groups. We further analyzed stimulation-induced changes in connectivity by calculating the difference between functional connectivity scores in the first and last MRI sessions per each ROI pair, on individual subjects ($\Delta$ rs-FC=rs-FC(S2)–rs-FC(S1)). These measures indicate the stimulation-induced changes in connectivity, normalized to baseline.

Throughout analysis, Levene's test was used to test for normality and homogeneity of variance, and Brown-Forsythe correction was used to correct the F-ratio and degrees of freedom when necessary. Figures were made using MATLAB and R (ggplot2, CRAN; *Patil, 2018*).

Finally, to explore the relationship between functional connectivity within the DVAN and behavioral performance, we computed the linear regression between using two independent individual measures of behavioral improvement (accuracy changes pre-post stimulation and accuracy changes during training) against individual measures of stimulation-induced changes in rs-FC. Only those subjects that were included in both behavioral and FC analysis were incorporated in the linear model analysis (a total of 26 subjects per each linear regression model). Using Pearson's correlation coefficient, we hence calculated the correlation between behavioral changes and brain activity modulation.

We also computed functional connectivity on resting-state data of 10 DMN ROIs defined based on previous imaging studies (*Buckner et al., 2008*; *Buckner et al., 2008*; *Crittenden et al., 2015*; *Andrews-Hanna et al., 2010*): the ventral medial prefrontal cortex, the dorsal prefrontal cortex, the left posterior cingulate cortex, the left lateral temporal cortex, the bilateral inferior parietal lobe (IPL), the bilateral parahippocampal cortex, and bilateral hippocampal formation (HF+). Coordinates in MNI space were mapped onto Talairach space using the 'MNI-to-Talairach converter with Brodmann Areas tool' included in the Yale BioImage Suite Package (*Papademetris et al., 2005*; *Papademetris et al., 2005*). We followed the same procedure used for the DVAN analysis to extract the time series within each ROIs, the computation of Pearson's r correlation coefficient and subsequent statistical analysis as for the DVAN. Finally, we compared the mean change in FC measured within the DVAN and the DMN. A two-way ANOVA was conducted with independent factors being network effects of stimulation condition and network on FC modulation to evaluate whether there was a difference in FC modulation within the two networks depending on stimulation conditions.

## Task-evoked brain activity

We analyzed BOLD signal changes in response to the OD and the TOJ task in the pre-stimulation session, before any manipulation (e.g., stimulation and training) could potentially change the brain response to the task. Therefore, we included all subjects in the analysis within one group (N=30, 2 subjects were excluded due to data motion contamination recorded during task-runs; leaving 28 subjects) and we observed the brain areas significantly activated while performing the two tasks. Pre-processing of task-related functional scans was performed using BrainVoyagerQX (Brain Innovations Inc, Maastricht, The Netherlands) and included the elimination of the first six volumes due to non-steady-state magnetization effects. Functional data were motion-corrected (3D motion correction) and adjusted for temporal fluctuation in intensity over time (linear-trend removal and mean intensity adjustment). Runs with head motion exceeding 3 mm in any given dimension and direction were discarded. Functional scans were initially co-registered using automatic BV procedures and then manually adjusted to each individual's high-resolution anatomical scan, then normalized into standardized Talairach coordinates (*Talairach and Tournoux, 1988*). BOLD activation was analyzed

**Table 2.** Clusters peak activation.

Peak activation coordinates of significant voxel clusters for the OD task are reported in Talairach space with relative number of voxels included in each cluster. Talairach Client was used to identy related anatomical/functional brain location and associated Broadmann Areas.

| | | | | | | OD | | |
|---|---|---|---|---|---|---|---|---|
| Cluster | X coor | Y coor | Z coor | No. of Voxels | Hemisphere | | Brain area | Broadmann area |
| 1 | 29 | –44 | 35 | 10,156 | Right cerebrum | Parietal lobe | VisuoMotor area | Broadmann area 7 |
| 2 | 47 | 1 | 21 | 714 | Right cerebrum | Frontal lobe | Inferior frontal gyrus | Brodmann area 9 |
| 3 | 35 | –59 | –24 | 4067 | Right cerebellum | Cerebellum | Culmen | * |
| 4 | 35 | 37 | 21 | 2337 | Right cerebrum | Frontal lobe | Middle frontal gyrus | Brodmann area 10 |
| 5 | 26 | –5 | 51 | 1203 | Right cerebrum | Frontal lobe | Middle frontal gyrus | Brodmann area 6 |
| 6 | 11 | –29 | 15 | 862 | Right cerebrum | Sub-lobar | Thalamus | Pulvinar |
| 7 | –4 | 7 | 48 | 6029 | Left cerebrum | Frontal lobe | Superior frontal gyrus | Brodmann area 6 |
| 8 | –43 | –50 | –15 | 11,400 | Left cerebrum | Temporal lobe | Fusiform gyrus | Brodmann area 37 |
| 9 | –16 | –5 | 18 | 470 | Left cerebrum | Sub-lobar | Caudate | Caudate body |
| 10 | –40 | –8 | 39 | 19,818 | Left cerebrum | Frontal lobe | Precentral gyrus | Brodmann area 6 |
| 11 | –40 | –38 | 30 | 21,367 | Left cerebrum | Parietal lobe | Supramarginal gyrus | Brodmann area 40 |
| 12 | –31 | 13 | 9 | 569 | Left cerebrum | Sub-lobar | Insula | Brodmann area 13 |
| 13 | –58 | –17 | 18 | 327 | Left cerebrum | Parietal lobe | Postcentral gyrus | Brodmann area 43 |

using a deconvolution GLM that included predictors for each volume (sampled at 2 s), each task-condition, extending 20 s following the onset of a trial. Statistical contrasts were estimated from the four volumes surrounding the peak of the hemodynamic response (4 s post-stimulus onset) for each voxel. In the first-level analysis, a GLM regression model was performed to fit data for each subject. Individual subject data were then combined into a deconvolution GLM in which beta estimates on OD and TOJ trials were contrasted with estimates for baseline activity. This contrast identified brain areas associated with OD task performance and TOJ performance, with statistical maps significance thresholded with a false discovery rate of 0.05 (FDR corrected). *Figure 8A* shows the results of the second-level deconvolution GLM analysis for significant BOLD response for the OD and TOJ tasks. The positive brain activations evoked by the two tasks largely overlap, with significant activity peaks over the parietal lobes (bilateral superior and IPLs), and on more anterior areas of the brain (including

**Table 3.** Peak activation coordinates of significant voxel clusters and the TOJ task are reported in Talairach space with relative number of voxels included in each cluster.

Talairach Client was used to identy related anatomical/functional brain location and associated Broadmann areas.

| | | | | | | TOJ | | |
|---|---|---|---|---|---|---|---|---|
| Cluster | X coor | Y coor | Z coor | | Hemisphere | | Brain area | Broadmann area |
| 1 | 47 | 1 | 21 | 1499 | Right cerebrum | Frontal lobe | Inferior frontal gyrus | Brodmann area 9 |
| 2 | 29 | –47 | 33 | 8339 | Right cerebrum | Parietal lobe | VisuoMotor area | Broadmann area 7 |
| 3 | 29 | –65 | –27 | 2278 | Right cerebellum | Posterior lobe | Pyramis | * |
| 4 | 35 | 37 | 21 | 2564 | Right cerebrum | Frontal lobe | Middle frontal gyrus | Brodmann area 10 |
| 5 | 11 | –29 | 15 | 359 | Right cerebrum | Sub-lobar | Thalamus | Pulvinar |
| 6 | –7 | 7 | 48 | 6975 | Left cerebrum | Frontal lobe | Medial frontal gyrus | Brodmann area 6 |
| 7 | –19 | –41 | 6 | 6430 | Left cerebrum | Limbic lobe | Parahippocampal gyrus | Brodmann area 30 |
| 8 | –40 | –8 | 39 | 23,304 | Left cerebrum | Frontal lobe | Precentral gyrus | Brodmann area 6 |
| 9 | –43 | –41 | 33 | 19,411 | Left cerebrum | Parietal lobe | Supramarginal gyrus | Brodmann area 40 |

FEFs bilateral areas). Details of the statistical maps with Talairach coordinates are reported in the table below (*Table 2* for OD and *Table 3* for the TOJ contrast, respectively). The Talairach daemon ( talairach.org) was used to identify the associated anatomical/functional brain regions and relative Broadmann areas (*Lancaster et al., 2000*; *Lancaster et al., 1997*).

## Acknowledgements

The present study was funded by the Autonomous Province of Trento, Call 'Grandi Progetti 2012,' project 'Characterizing and improving brain mechanisms of attention – ATTEND' (FC, LB) and The Blavatnik Family Foundation (Blavatnik Sensory Disorders Research Award) to LB; EG was supported by the National Science Foundation BCS0748314. Any opinions, findings, and conclusions, or recommendations expressed in this material are those of the author(s) and do not necessarily reflect the views of the National Science Foundation.

## Additional information

### Funding

| Funder | Grant reference number | Author |
|---|---|---|
| Provincia Autonoma di Trento | Grandi Progetti | Lorella Battelli |
| National Science Foundation | BCS0748314 | Emily Grossman |
| Blavatnik Family Foundation | Blavatnik Sensory Disorders Research Award | Lorella Battelli |

The funders had no role in study design, data collection and interpretation, or the decision to submit the work for publication.

### Author contributions

Federica Contò, Conceptualization, Data curation, Formal analysis, Investigation, Methodology, Writing – original draft, Writing – review and editing; Grace Edwards, Data curation, Formal analysis, Supervision, Writing – review and editing; Sarah Tyler, Conceptualization, Data curation, Methodology; Danielle Parrott, Investigation; Emily Grossman, Data curation, Methodology, Supervision, Writing – review and editing; Lorella Battelli, Conceptualization, Methodology, Supervision, Writing – review and editing

### Author ORCIDs

Federica Contò  http://orcid.org/0000-0003-1760-4110
Lorella Battelli  http://orcid.org/0000-0001-9981-4028

### Ethics

Human subjects: The study was approved by the ethical committee of the University of Trento.

### Decision letter and Author response

Decision letter https://doi.org/10.7554/eLife.63782.sa1
Author response https://doi.org/10.7554/eLife.63782.sa2

## Additional files

### Supplementary files
• Transparent reporting form

### Data availability

All data generated and analyzed during this study are available for review at the links reported in the Transparent Reporting File (Conto_transparent_reporting_resubmission). Moreover, a data structure explanation has also been provided (file name: Conto;_DataStuctureExplanation_resubmission).

Data have been deposited to Zenodo and are available at the following DOIs: https://doi.org/10.5281/zenodo.4634008 https://doi.org/10.5281/zenodo.5558975, https://doi.org/10.5281/zenodo.4621644.

The following dataset was generated:

| Author(s) | Year | Dataset title | Dataset URL | Database and Identifier |
|---|---|---|---|---|
| Contò F, Edwards G, Tyler S, Parrott D, Grossman ED, Battelli L | 2021 | Behavioral data collected during pre_post sessions | https://doi.org/10.5281/zenodo.4621644 | Zenodo, 10.5281/zenodo.4621644 |
| Contò F, Edwards G, Tyler S, Parrott D, Grossman ED, Battelli L | 2021 | Behavioral data collected during training | https://doi.org/10.5281/zenodo.4634008 | Zenodo, 10.5281/zenodo.4634008 |
| Contò F, Edwards G, Tyler S, Parrott D, Grossman ED, Battelli L | 2021 | fMRI data collected during prepost sessions | https://doi.org/10.5281/zenodo.5558975 | Zenodo, 10.5281/zenodo.5558975 |

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

## Appendix

### Resting-state default mode network FC analysis

We examined whether the effect of multi-session hf-tRNS and training on resting-state functional connectivity (rs-FC) was specific for the targeted DVAN by analyzing changes in the connectivity of a different network, the default mode network (DMN). This network comprises a series of large regions distributed on the cortex and shows increased activity levels during rest or when the subjects are not involved in an active goal-oriented state (*Raichle et al., 2001*; *Greicius et al., 2003*; *Buckner et al., 2008*; *Andrews-Hanna et al., 2010*; *Braga and Buckner, 2017*). We hypothesized that functional connectivity within the DMN would not be modulated by the stimulation protocol we used.

To test whether stimulation coupled with training had an effect on this control network, we run the same analysis performed for the DVAN. First, the mean connectivity per subject within the DMN was calculated by averaging the FC scores of the 45 pairs, and then averaged across subjects in each group. Next, we analyzed whether there was an effect of stimulation on FC scores of this network by performing a mixed repeated measures ANOVA with fMRI session ($n = 2$; pre- vs. post-stimulation session) as within and stimulation site as between-subjects factor ($n = 3$; Parietal, hMT+, and Sham). This analysis did not show any significant effect of stimulation site ($F_{(2,27)}=0.09$, p=0.91), nor of time ($F_{(1,27)}=058$, p=0.45). Importantly, the interaction between stimulation condition and session was not significant ($F_{(2,27)}=0.20$, p=0.81), indicating that the rs-FC scores within the DMN did not change over time depending on the stimulation group (*Appendix 1—figure 1*).

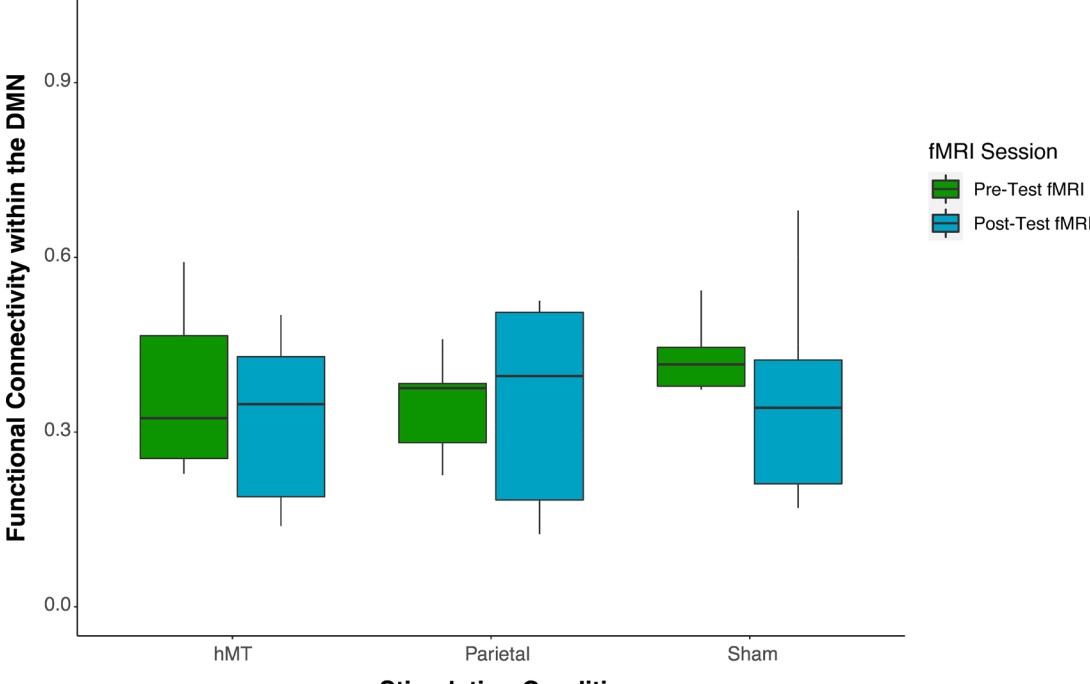

**Appendix 1—figure 1.** Pre- and post-stimulation functional connectivity (FC) changes within the default mode network (DMN). Overall mean FC of the DMN averaged per stimulation condition (Parietal, hMT+, and Sham) and per fMRI session (pre- and post-stimulation sessions).

To further test whether FC within the DMN was affected by stimulation condition, we compared the rs-FC of the three stimulation groups at each time point (pre- and post-stimulation day) by performing two one-way ANOVAs, one per session (pre- and post-stimulation session). These analyses found no significant difference in connectivity between stimulation groups in the pre-stimulation session ($F_{(2, 27)}=0.01$, p=0.82) nor in the post-stimulation sessions ($F_{(2, 27)}=0.10$, p=0.90), indicating that connectivity scores within this network did not differ at baseline (prior to stimulation) and, importantly, did not change after the stimulation sessions.

We performed a further analysis to test whether stimulation affected the connectivity modulation within the DMN. As performed for the DVAN, we calculated the difference in FC scores between the pre- and post-stimulation sessions to measure the delta scores (modulation connectivity) for each condition. A one-way ANOVA on delta FC scores showed there was no significant difference between stimulation conditions (F(2, 27)=0.20, p=0.81).

We then run the second set of control analysis to investigate the modulation of the DMN by taking into consideration all ROIs pairs as single components of the network per subject (n=45 ROIs to ROIs pairs per subject), instead of using one averaged correlation value for the whole network as in the previous analysis (as performed for the DVAN). We first performed a one-way ANOVA to compare FC between groups prior to stimulation and we found no significant difference (F(2, 1298.682)=0.700, p=0.497, Brown-Forsythe p-value and degrees of freedom corrected), indicating that FC did not differ between stimulation groups at baseline (pre-stimulation session). We then performed a mixed repeated measures ANOVA to test the effect of the stimulation site and session on all correlation values within the DMN. This analysis revealed no main effect of session on FC scores (F(1, 1347)=2.4, p=0.116), nor of stimulation group (F(2, 1347)=0.347, p=0.707). Importantly, the interaction between stimulation group and session was also not significant (F(2, 1347)=0.853, p=0.426), indicating the absence of a significant difference between rs-FC scores on the two sessions depending on the stimulation condition.

Finally, we compared rs-FC on all single components (pairs) of the post-test stimulation DMN by performing a one-way ANOVA. Because Levene's test indicated that the assumption of homogeneity of variances was violated, the Brown-Forsythe F-ratio is reported. This analysis revealed non-significant effect of stimulation condition on post-stimulation FC scores (F(2, 1297.89)=0.437, p=0.646), which indicates that FC did not differ between groups on this session (*Appendix 1—figure 2*).

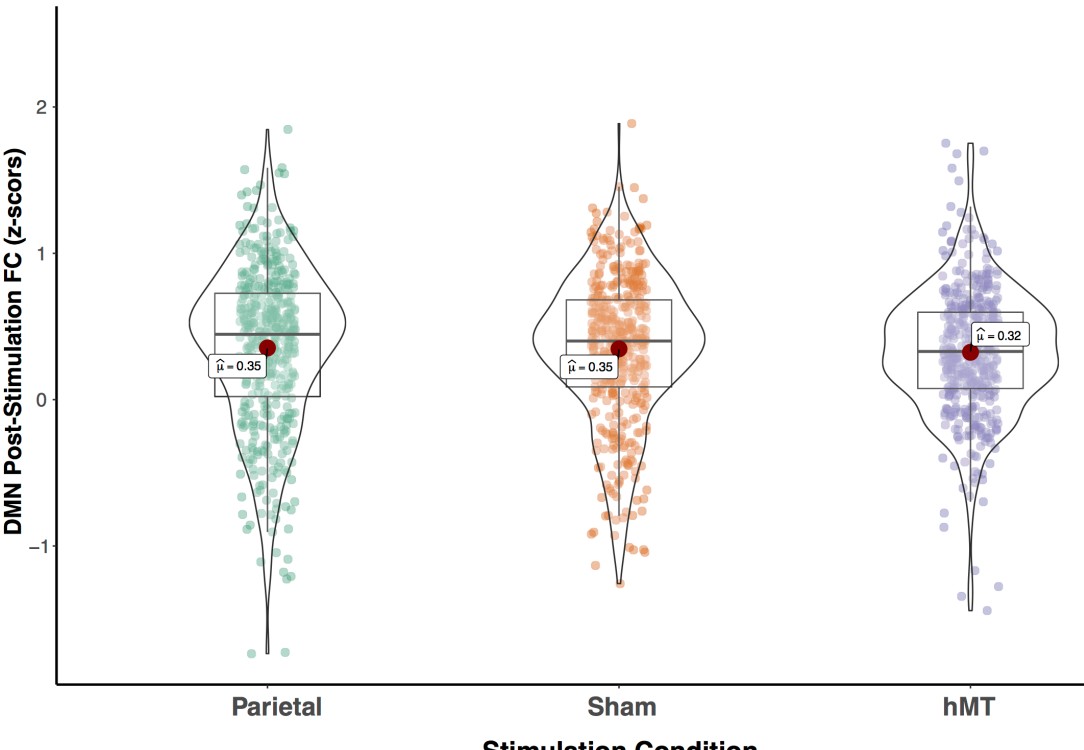

**Appendix 1—figure 2.** Functional connectivity within the DMN after multi-session hf-tRNS coupled with training (post-stimulation session). Correlations of all ROI pairs (z-scores) constituting the DMN are represented separately for the three stimulation conditions (Parietal, Sham, and hMT+). No differences were found between stimulation conditions. DMN, default mode network; ROI, region of interest; tRNS, transcranial random noise stimulation

