## [Editor Report]

Conto et al. applied transcranial random noise stimulation to the parietal cortex during performance of an attention task, examining if multiple sessions would result in a cumulative increase in performance. The results showed that tRNS to parietal, but not a ventral-lateral temporal site, increased performance on an orientation discrimination task, but not a temporal order judgement task. Using resting-state functional magnetic resonance imaging and a functional connectivity (FC) analysis, tRNS was shown to increase FC within the dorsal and ventral attention network. Overall, an increase in FC within the DVAN positively correlated with changes in behavioral performance (collapsing across all stimulation conditions). Overall, the experimental design and results are compelling. These findings further our understanding of how combining non-invasive brain stimulation with behavioral training can lead to enhanced neural plasticity that supports more effective learning.

---

## [Decision Letter]

Thank you for submitting your article "Attention network modulation via tRNS correlates with attention gain" for consideration by *eLife*. Your article has been reviewed by 3 peer reviewers, including Taraz Lee as the Reviewing Editor and Reviewer #1, and the evaluation has been overseen Rich Ivry as the Senior Editor.

The reviewers have discussed the reviews with one another and the Reviewing Editor has drafted this decision to help you prepare a revised submission.

As the editors have judged that your manuscript is of interest, but as described below that additional analyses are required before it is published, we would like to draw your attention to changes in our revision policy that we have made in response to COVID-19 (https://elifesciences.org/articles/57162). First, because many researchers have temporarily lost access to the labs, we will give authors as much time as they need to submit revised manuscripts. We are also offering, if you choose, to post the manuscript to bioRxiv (if it is not already there) along with this decision letter and a formal designation that the manuscript is "in revision at *eLife*". Please let us know if you would like to pursue this option. (If your work is more suitable for medRxiv, you will need to post the preprint yourself, as the mechanisms for us to do so are still in development.)

Summary:

Conto et al. applied transcranial random noise stimulation to the parietal cortex during performance of an attention task, examining if multiple sessions would result in a cumulative increase in performance. The results showed that tRNS to parietal, but not a ventral-lateral temporal site, increased performance on an orientation discrimination task, but not a temporal order judgement task. Using resting-state functional magnetic resonance imaging and a functional connectivity (FC) analysis, tRNS was shown to increase FC within the dorsal and ventral attention network. Overall, an increase in FC within the DVAN positively correlated with changes in behavioral performance (collapsing across all stimulation conditions). Overall, the experimental design and results are compelling. However, there are several concerns that need to be addressed before publication can be recommended.

Essential revisions:

1. Multiple reviewers felt the flow of the paper from the introduction to the results is quite jarring given that there is very little explanation of the tasks or the overall design of the experiment. What is the orientation discrimination task, for example? When do the resting state scans occur? Figure 7 is quite informative, "absolutely essential" in the words of one reviewer, and really needs to be the first figure the reader sees. The authors should include a paragraph at the start of the results that elaborates the full experimental design and motivation for including each element. Additionally, please explicitly acknowledge the time at which the resting-state fMRI sessions were acquired in Figure 7. It seems that task-based fMRI was collected on the same days as the resting state scans. Did the resting scans occur before or after task performance? Given that prior studies have reported that tasks can alter subsequent resting state connectivity, it seems important to know whether tRNS altered baseline connectivity or rather post-task connectivity. Please use the same terminology in Figure 2 "day 1 or day 6" or "pre-test" and "post-test". The authors switch terminology occasionally between figures, and with an already complex experimental design, this adds undue complexity.

2. Why is data from the training sessions not reported? It seems important to see whether stimulation improved on-line performance during training to properly interpret the results. The authors should report these data (ideally with a figure). On a related note, it is unclear why the authors chose to show the dependent measure is that is reported in Figure 1. Given that what was actually analyzed was the pre-test and post-test accuracy scores, could these not be plotted instead of a % improvement score (i.e. similar to Figure 2)?

3. The authors claim that stimulation to IPS alters DVAN connectivity specifically and that this change in connectivity is what drives behavioral improvements. However, they do not assess changes in network connectivity within any other networks. Perhaps IPS stimulation leads to more widespread connectivity changes that also have an effect. One option is to investigate a network that is not involved in the attention task, e.g. the default mode network. Functional connectivity within this network should not be modulated by stimulation. This control analysis would lend further evidence to confirm the specificity of the stimulation effects. In addition, the authors could include an analysis of the function network targeted by hMT+. Perhaps functional connectivity within this network increases from hMT+, but does not translate into a benefit into performance (no correlation between FC and performance). Alternatively, stimulation to hMT+ may not modulate the functional network because that network is not active during the task, providing additional evidence to the state-dependent effects of tRNS.

4. The difference in electrode placement for parietal and hMT+ stimulation is subtle and thus speaks to the specificity of the effects. However, the differential effect of stimulation on the brain due to such a slight adjustment of electrode placement should be elaborated. Please include electric field models (e.g. ROAST) for the two electrode montages. The effect of parietal stimulation on the DVAN should be more apparent from these electric field models.

5. In the discussion, the authors reveal that hMT+ was targeted as a region that may play a role in temporal attention and the experiment was thus designed to probe this cognitive process. However, in the introduction this motivation for the experimental design is not revealed but instead referred to as "an active control". The authors can be more forthcoming in the introduction about this hypothesized link and that it did not pan out. This is acceptable and hMT+ can still serve as an active control.

6. There is very little information provided about the pairwise functional connectivity analysis (starting on line 193). Is each pair treated independently in this analysis? It seems as though the analysis conducted does not take into account within-participant variance given the DF reported? More information needs to be provided in (perhaps in the Methods) about this analysis.

7. The authors are very underpowered to conduct a brain-behavior correlation and this analysis does not seem very informative given that there are 3 different groups. Given that there doesn't seem to be much of a relationship between behavioral and functional connectivity changes in the sham and hMT+ group, it seems this is all driven by the parietal group and has already been shown in the prior analyses. The authors should consider removing this analysis or at least providing more motivation behind the need for this analysis

[Editors' note: further revisions were suggested prior to acceptance, as described below.]

Thank you for resubmitting your work entitled "Attention network modulation via tRNS correlates with attention gain" for further consideration by *eLife*. Your revised article has been evaluated by Richard Ivry (Senior Editor) and a Reviewing Editor.

The manuscript has been improved but there are some remaining issues that need to be addressed, as outlined below:

– Although they were enthusiastic about the result multiple reviewers commented on concerns regarding a small sample size and an inflated effect size. Although additional stimulation-behavioral data to more properly estimate the effect size would be welcome, the authors need to at least: report effect sizes, state how power was calculated for the study, and caution that the effect sizes observed are likely to be somewhat inflated given that they were conditioned on a significant result.

– There was still concern about the electric field models presented in the paper and their relationship with the activity in the stimulation sites that is driven by the tasks at hand. In consultation, one reviewer remarked, "The SimNIBS e-field models are presented in a raw form that in my experiences is only shown as a challenge getting the software to display the e-field on the cortical surface. I suggest that the authors run their e-field models in the ROAST toolbox. As presented, the e-fields are not really interpretable." The reviewers felt it would be important to compare whole-brain maps of task-related activity with accurate electric field models to determine whether the effects of stimulation are due to more effective targeting of task-active regions.

– Several issues with clarity and claims made in the discussion need to be resolved. Please see the specific comments made by reviewers #4 and #5 below.

*Reviewer #4:*

N=10 / group is fairly small. Please report effect sizes. How was power calculated for the study?

Unclear why data is represented inconsistently (i.e., post-stimulation scores at times and difference between FC scores pre- and post-test sessions at other times). Does post-stimulation refer to the change from baseline or just data from the post-session alone? The difference between FC scores pre- and post-test sessions is more informative and should be what is reported in the manuscript.

Scatter plots should be removed.

*Reviewer #5:*

This study shows a very striking effect of parietal tRNS on object discrimination and corresponding large changes in functional connectivity. Such data are sure to be of interest to numerous researchers due to their practical significance. That said, I have statistical concerns and am left wanting more mechanistic insight. Ultimately, I think more data and analyses will be needed to make a convincing enough case to the readership of this outlet.

1) The effect of parietal tRNS on object discrimination is truly massive. The tasks are titrated at the start to set performance at 50%. While performance remains the same following temporal and sham tRNS, parietal tRNS improves performance by 30%. Given this finding, I am rather convinced that parietal tRNS improves object discrimination. What I am not convinced of is that the effect size is as massive as is being depicted. Given that the main contribution of this study as it is current written is practical, having a reasonable estimate of effect size is important for applied work that will seek to build on this striking finding. However, given the small sample size (10 participants per group), and effect size inflation that occurs when conditionalizing on significance, especially in small samples, I suspect that the true effect size is substantially smaller than has been depicted. Unfortunately, I do not see a good way to get a better estimate of the true effect size without acquiring more data. The upshot is that new data would not require fMRI and may not even require control groups. A bare minimum would be to collect another group of 10 participants receiving parietal tRNS and estimate effect size based on their improvement. I think that such data collection would be a low cost effort, and the sample size is small enough to potentially be feasible amidst pandemic restrictions. Note that since the new sample would be used to estimate effect size, there need not be statistically significant improvement in the new sample to have utility.

2) These data beg the question of why parietal tRNS produces such a dramatic effect, but temporal tRNS does not. That is, one would like to have some mechanistic insight into why parietal tRNS is so effective, at least in the object discrimination task. The connectivity data do not, in their present form, address this question. They merely provide another metric of change without begetting further causal inference/insight. Fortunately, the authors have additional task fMRI data that could potential shed some light on the issue. For example, one might ask whether the extent to which a given task activates the stimulated areas is associated with changes in connectivity and/or behavior. Alternatively, the network position of the parietal target either generally or specific to a given task may be associated with its effective change in connectivity and/or behavior. Given that the authors have effects that are specific not only to target, but also differ as a function of task, I feel as though these data could be leveraged to provide substantial insights into what factors are critical for an effective intervention.

Relatedly, the authors offer conclusions in the discussion regarding what constitutes a successful intervention, but they are either unsubstantiated by the present data, or unclear. For example, they state on page 20, lines 439-441:

"It is the combination of a challenging task with concurrent brain stimulation over the relevant cortical circuits that facilitates fast learning, a paradigm amenable to promote plasticity."

Similar sentiments are echoed on page 20, lines 451-453. Absent examination of task data, the relevant cortical circuits for the studied tasks are unclear. Consider that in Figure 9, the temporal targeted tRNS seems to induce the most stimulation in occipital areas. I would expect occipital cortex to be extremely relevant for the studied tasks, yet stimulation over those areas did not improve performance. Moreover, the importance of stimulation concurrent with tasks cannot be concluded from these data since there were no conditions that had no tasks. The differential performance across the studied tasks may potentially be leveraged here, but at the moment, this conclusion does not follow from the data. Since the authors have data that can help speak to some of these issues, it seems pertinent to use them.

[Editors' note: further revisions were suggested prior to acceptance, as described below.]

Thank you for resubmitting your work entitled "Attention network modulation via tRNS correlates with attention gain" for further consideration by *eLife*. Your revised article has been evaluated by Richard Ivry (Senior Editor) and a Reviewing Editor.

The manuscript has been greatly improved but there are some remaining issues that need to be addressed, as outlined below. Though these revisions are essential, the manuscript will not be sent out for another round of external reviews and can be handled editorially.

1. While the Bayesian analyses are nice, they do not address the point that the effect sizes are likely to be inflated given the sample size. The issue is that when sample size is small and reporting of results is conditioned on statistical significance (e.g. p< 0.05) , effect size will be inflated. This needs to be acknowledged in the discussion. See, for example, the reference below for more discussion of inflated effect sizes in studies with small samples.

Yarkoni, T. (2009). Big correlations in little studies: Inflated fMRI correlations reflect low statistical power-Commentary on Vul et al. (2009). Perspectives on psychological science, 4(3), 294-298.

2. Please report the analyses showing that the number of frames censored were similar across sessions in the manuscript.

3. The task-based univariate results are quite illuminating. It would seem pertinent to mention in the manuscript that both tasks activate IPS in the discussion of why effects were observed for the OD task but not the TOJ task.

---

## [Author Response]

Essential revisions:1. Multiple reviewers felt the flow of the paper from the introduction to the results is quite jarring given that there is very little explanation of the tasks or the overall design of the experiment. What is the orientation discrimination task, for example? When do the resting state scans occur? Figure 7 is quite informative, "absolutely essential" in the words of one reviewer, and really needs to be the first figure the reader sees. The authors should include a paragraph at the start of the results that elaborates the full experimental design and motivation for including each element. Additionally, please explicitly acknowledge the time at which the resting-state fMRI sessions were acquired in Figure 7. It seems that task-based fMRI was collected on the same days as the resting state scans. Did the resting scans occur before or after task performance? Given that prior studies have reported that tasks can alter subsequent resting state connectivity, it seems important to know whether tRNS altered baseline connectivity or rather post-task connectivity. Please use the same terminology in Figure 2 "day 1 or day 6" or "pre-test" and "post-test". The authors switch terminology occasionally between figures, and with an already complex experimental design, this adds undue complexity.

Thank you for the positive and constructive comments, we have moved Figure 7 within the Results section, and it is now Figure 1. As suggested, at the beginning of the Results section, we have added a paragraph briefly summarizing the experimental procedure, the tasks and motivation. More details to explain the time course of the experiment have now been added to the Figure legend. We have revised the manuscript and used the same terminology to indicate pre and post-test sessions throughout the text and in figure caption.

Resting state scans were collected on the same day and during the same task-based fMRI scanning session (see new text in the *Neuroimaging Procedure* section). Because we followed the same acquisition order for each participant and each fMRI session (pre- post-test sessions), if task performance had any effect on resting state connectivity, its potential impact would be equivalent for each participant and condition. Importantly, to assess whether task performance differently influenced resting state connectivity and yielded differences between groups at baseline, we performed statistical analysis on the pre-test fMRI session connectivity data (Day2 in Figure 1A). We compared resting state functional connectivity within the DVAN for the three stimulation conditions and we did not find any significant difference in pre-stimulation FC scores between the three groups (see second paragraph of ‘Stimulation and learning-dependent changes in functional connectivity’ in Results section), indicating that connectivity did not differ between groups at baseline. Thus, the changes we found on Day 7 (Figure 1A) were solely due to the stimulation and training protocol. Moreover, we analyzed the modulation of connectivity scores by normalizing post-test connectivity to pre-test connectivity (∆=FC(S2) – FC(S1)), so that potential differences at baseline were accounted for (Figure 6, left panel).

2. Why is data from the training sessions not reported? It seems important to see whether stimulation improved on-line performance during training to properly interpret the results. The authors should report these data (ideally with a figure). On a related note, it is unclear why the authors chose to show the dependent measure is that is reported in Figure 1. Given that what was actually analyzed was the pre-test and post-test accuracy scores, could these not be plotted instead of a % improvement score (i.e. similar to Figure 2)?

Following the reviewers’ suggestion, we have now reported the training data and analysis in the revised manuscript (subsection ‘Time Course of Learning’ in Results section), including figures showing percentage improvement (Figure 2C and 2D). We initially wanted to focus on the same-session effects upon behavior and functional connectivity, specifically on the pre-test and post-test sessions (Day2 and Day 7 in Figure 1A, respectively). However, we acknowledge that the time course of training reveals how long it took for the learning to build up and become significantly different from the control conditions, where we did not see any learning. The new analysis performed on the training data confirmed the main result found in the pre- vs post-test analysis. The selective improvement for the visuospatial task (Orientation Discrimination, OD) built up and consolidated across sessions, while the temporal order judgment (TOJ) task did not change, and this was true for the parietal group only. We have now modified the results and Discussion sections to include training data. Figure 2 (Behavioral performance on the OD and the TOJ task during and following stimulation and training) has been modified and has now four panels reporting the box plots for behavioral improvement between pre- and post-test sessions for the OD (Figure 2A) and the TOJ task (Figure 2B), and the time course of the effect across the four training sessions for the OD (2C) and the TOJ (2D) task, separately.

To respond to the second point raised by the reviewers, in Figure 2 we show the dependent measure as the difference in performance between pre- and post-test accuracy for each task because at baseline (pre-test) the one-way ANOVA did not show any significant difference in performance between groups, while it revealed strong significant difference in the post-test. Normalized data directly represents the change in performance for each stimulation condition against baseline performance (Figure 2A and Figure 2B).

3. The authors claim that stimulation to IPS alters DVAN connectivity specifically and that this change in connectivity is what drives behavioral improvements. However, they do not assess changes in network connectivity within any other networks. Perhaps IPS stimulation leads to more widespread connectivity changes that also have an effect. One option is to investigate a network that is not involved in the attention task, e.g. the default mode network. Functional connectivity within this network should not be modulated by stimulation. This control analysis would lend further evidence to confirm the specificity of the stimulation effects. In addition, the authors could include an analysis of the function network targeted by hMT+. Perhaps functional connectivity within this network increases from hMT+, but does not translate into a benefit into performance (no correlation between FC and performance). Alternatively, stimulation to hMT+ may not modulate the functional network because that network is not active during the task, providing additional evidence to the state-dependent effects of tRNS.

Following reviewers’ suggestion, we conducted a number of control analysis to examine functional connectivity changes in the Default Mode Network (DMN), following training and stimulation. This network comprises a number of functionally interconnected brain regions whose activity increases during rest or when the person is not involved in an active goal-oriented state (Greicious et al., 2003; Buckner et al., 2008; Andrews-Hanna et al., 2010; Braga, et al., 2017; Raichle et al., 2001; Parker and Razlighi, 2019). Because this network was not targeted by our stimulation protocol and it is not directly involved in attention tasks, we hypothesized that the connectivity within its nodes would not be modulated by our protocol. We performed the same analysis we conducted for the DVAN rs-FC to test whether changes in connectivity were specific for the DVAN, or whether stimulation induced more widespread changes in the brain, affecting other networks, like the DMN.

We have now reported all the analysis and results relative to the DMN in the manuscript and Appendix (Appendix, section “Default Mode Network Analysis”). Briefly, the control analyses found no changes in connectivity within the DMN, indicating that our stimulation-induced changes were selective for the DVAN.

In addition to the lack of DMN modulation, we also demonstrate the selectivity of our stimulation-training protocol by the modulation of behavior and rs-FC following stimulation to the Parietal regions only. Our results suggest that when bilateral tRNS is applied over brain regions critical for the attention network (parietal areas), it induces propagated connectivity changes within this network and supports related behavior. However, stimulation to hMT+ areas which are involved, but not critical for attentional processes, does not induce attentional network nor behavior changes. Interestingly, in a previous study from our group (Herpich et al., 2019), we showed that when we targeted the IPS with tRNS while subjects were training on a motion coherence task (a task hMT+ is very sensitive to, as shown by imaging and physiological studies; Tootell et al., 1995; Born and Bradley, 2005), we found no effect of stimulation. Thus, we believe our past and current data indeed provide evidence of site-specific effects of tRNS.

Together, the control analysis on the DMN and the selectivity of the site-specific stimulation effect on functional connectivity, provide strong evidence for a selective change in connectivity within the Dorsal and Ventral Attention network. Our findings do not support the hypothesis of tRNS causing a general unspecific effect on the brain (or behavior), discussed further below.

4. The difference in electrode placement for parietal and hMT+ stimulation is subtle and thus speaks to the specificity of the effects. However, the differential effect of stimulation on the brain due to such a slight adjustment of electrode placement should be elaborated. Please include electric field models (e.g. ROAST) for the two electrode montages. The effect of parietal stimulation on the DVAN should be more apparent from these electric field models.

We agree with the reviewer, while the distance between electrode location for hMT+ and Parietal is small, we still found a strong difference in the behavioral and imaging results, a clear indication of the specificity of the effect. We believe one of the reasons is that we carefully chose and programmed a task that was well known to increase activity within the attention-related parietal areas. We then maximally recruited parietal areas through increasing task-difficulty via thresholding performance for each participant. Thresholding also equated task-difficulty across subjects, to ensure any changes we found were due to our manipulation (different stimulation conditions) and not to other variables.

As the reviewer suggested, we also performed a simulation of the electrical field current distribution in the head using SimNIBS (Saturnino et al., 2019; Thielscher et al., 2015) and included the results for each active stimulation setting (Parietal and hMT+) in “Stimulation Protocol” section, Figure 8. These electric field models aim to determine which brain areas are affected by different types of electrode montages (Saturnino et al., 2019), allowing a more careful evaluation of whether the electrode position and the corresponding current intensities are targeting the brain area we stimulated. However, single brain regions are not segregated and do not operate in isolation. Yet, they constantly interact with other regions in distributed brain networks (Grefkes and Fink, 2014; Yeo et al., 2011; Siegel et al., 2015). A limitation of these electric field models is their difficulty to determine whether a stimulation setting will influence the site of stimulation only, or whether it will produce a cascade of network-wide effects. Our results suggest that when bilateral tRNS is applied over parietal cortices which are critical brain regions for the attention network, it is able to induce a cascade of connectivity changes within this network (and support related behavior), while when stimulation is applied over a nearby but distinct brain region not crucial for attention, like hMT+, it does not induce attentional network nor behavior changes. Importantly, these results are consistent with previous studies that found different behavioral results on an attention task using the same stimulation settings (Tyler et al., 2018). Together, these results represent new findings for this stimulation technique, as they indicate that the effects of tRNS might have a higher spatial resolution than previously thought and points to the possibility to use these neuromodulation and cognitive training protocol to exert specific, long-lasting and distal effects.

5. In the discussion, the authors reveal that hMT+ was targeted as a region that may play a role in temporal attention and the experiment was thus designed to probe this cognitive process. However, in the introduction this motivation for the experimental design is not revealed but instead referred to as "an active control". The authors can be more forthcoming in the introduction about this hypothesized link and that it did not pan out. This is acceptable and hMT+ can still serve as an active control.

Thank you for pointing this out. We have now added a clearer motivation in the Introduction on our rationale for choosing hMT+ as an active control condition. Interestingly, tRNS over hMT+ had no effect in our cross-tasks training procedure. It is also remarkable that the temporal order judgment task was not affected by the stimulation. Many reasons could account for this result. However, as we speculate in the Discussion section, when the task is to detect an interval time difference between targets displaced across space (and hemifields), more ventral, inferior parietal areas might be directly involved (Battelli et al., 2007), areas that we did not stimulate in our experiment.

6. There is very little information provided about the pairwise functional connectivity analysis (starting on line 193). Is each pair treated independently in this analysis? It seems as though the analysis conducted does not take into account within-participant variance given the DF reported? More information needs to be provided in (perhaps in the Methods) about this analysis.

We revised the manuscript in response to reviewers’ comments. We have now added additional details in the ‘*Methods’* section, *‘Resting State fMRI data analysis’* subsection.

7. The authors are very underpowered to conduct a brain-behavior correlation and this analysis does not seem very informative given that there are 3 different groups. Given that there doesn't seem to be much of a relationship between behavioral and functional connectivity changes in the sham and hMT+ group, it seems this is all driven by the parietal group and has already been shown in the prior analyses. The authors should consider removing this analysis or at least providing more motivation behind the need for this analysis

We agree with the reviewer that our prior analyses already show that FC changes in the parietal group only. However, we believe the connectivity-behavior correlation analysis is critical to determine whether connectivity modulation within the attention network is related to performance changes, and answers the question of potential driving/causal relationship between stimulation-induced changes and facilitatory behavioral effects. In particular, we hypothesized that the magnitude of connectivity modulation induced by the stimulation protocol coupled with training could covary with behavioral improvements measures. These analyses are consistent with previous work that correlated levels of BOLD signal and behavior as a way to assess the function of a given brain region for a given performance, and to explore the relationship between brain dynamics patterns and behavior (Kang et al., 2018; Fatima et al., 2016; Wen et al., 2012; Hester et al., 2008; Rypma et al., 2002).

To further analyze the behavioral relevance of connectivity within the attention network, we now have incorporated two correlation analyses (see modified manuscript for details, subsection ‘Functional connectivity-behavior correlation’ in Results section). In particular we computed two linear regression models using individual measures of behavioral improvement (one for accuracy changes pre-post stimulation and one for accuracy changes during training) against individual measures of stimulation-induced changes in rs-FC. The results indicate that individual differences in performing the attention task both during and after training/stimulation can be related to individual differences in functional connectivity changes within the attention network.

We agree that our correlation analysis could be underpowered, a higher number of data points might increase the strength of its significance. However, we also believe that the results obtained are informative and useful to interpret our data, and mostly of the relationship between connectivity and behavior stimulation-induced effects. We were motivated to perform this further analysis and to include it in the manuscript by a number of published studies that used a similar number of subjects to look at brain-behavior correlation (Rosenberg et al., 2020; Sreenivasan et al., 2019; Frangou et al., 2018; Kang et al., 2018; Ficek et al., 2018; Wen et al., 2012;; Fatima et al., 2016; Baldassarre et al., 2012; Bonnelle et al., 2012; Fox et al., 2007; Young et al., 2014).

[Editors' note: further revisions were suggested prior to acceptance, as described below.]

The manuscript has been improved but there are some remaining issues that need to be addressed, as outlined below:– Although they were enthusiastic about the result multiple reviewers commented on concerns regarding a small sample size and an inflated effect size. Although additional stimulation-behavioral data to more properly estimate the effect size would be welcome, the authors need to at least: report effect sizes, state how power was calculated for the study, and caution that the effect sizes observed are likely to be somewhat inflated given that they were conditioned on a significant result.

Thank you for the positive comments, we have now updated the manuscript and reported the effect size for each statistical test, indicating the magnitude of the experimental effects. Reviewer 5 states that they are convinced that parietal tRNS improves object discrimination performance but expresses their concern about how “truly massive” the behavioral effect is in this sample. We have now conducted a series of additional analyses to rule out the possibility that our results were inflated. In particular, we ran a Bayes factor analysis which is an alternative statistical approach that provides an estimate of the amount of evidence present in the data, and it does so both for the research hypothesis and for the null hypothesis (Kruschke, 2011), thus overcoming some of the shortcomings of classical null-hypothesis significance testing and associated p-values (Johnson, 2013; Nuzzo, 2013; Jarosz and Wiley, 2014). Importantly, the Bayes factor analysis deals better with small sample sizes and helps distinguish between evidence for an unsuccessful stimulation protocol and inconclusive evidence (Biel and Friedrich, 2018). We calculated the Bayes factor as the ratio that contrasts the likelihood of the data fitting under the null hypothesis with the likelihood of the data fitting under the alternative hypothesis for the two main significant behavioral analyses, that is, for the repeated measure ANOVA and for the one-way ANOVA on the performance change between pre- and post-stimulation on the orientation discrimination task. Results are reported in Author response table 1. Using the interpretation of the Bayes Factors (BFs) suggested by Lee and Wagenmakers (2013), the estimated BFs calculated on our behavioral data (BF10 = 99.956 for the Bayesian one-way ANOVA, and BF10=83.962 for the main result of the Bayesian rm-ANOVA) indicate that there is very strong evidence in support of the alternative hypothesis (H1). In particular, the Bayesian one-way ANOVA indicates that the data were 99.956 times more likely to occur under the model including the effect of stimulation condition, compared to the model without this effect. In order to follow up on this result, we then compared each level of the dependent variable with post-hoc Bayes factor t-test. Parietal group vs Sham and Parietal vs hMT revealed posterior odds of 18.103 and 11.304, which indicates strong evidence in favor of the alternative hypothesis.

**Author response table 1. sa2table1:** Bayes Factor Analysis on Orientation Discrimination data. Table 1a and 1b reports the results of the BF analysis that tested the two models, null and Stimulation Condition (Condition). P(M) indicates the prior probabilities of each model to be equal (prior model odds). P(M|data) reports the updated probabilities having taken into consideration the data (posterior model probabilities); BFM indicates how much the data have changed the prior model odds. BF10 indicates the Bayes factors for each model (the BF10 for stimulation condition indicates how many times data are more likely to fall under the stimulation model, rather than the null model). Error % indicates the sensitivity to numerical fluctuations. Table 1c reports the results of the post-hoc Tests BF analysis. The posterior odds have been corrected for multiple testing by fixing to 0.5 the prior probability that the null hypothesis holds across all comparisons (Westfall et al. 1997). Individual comparisons are based on the default t-test with a Cauchy (0, r = 1/sqrt(2)) prior. The "U" in the Bayes factor denotes that it is uncorrected.

Table 1a					
Bayesian Repeated Measures ANOVA					
Data Input: OD pre, and post-stimulation performance					
Model Comparison					
**Models**	**P(M)**	**P(MIdata)**	**BF** _ **m** _	**BF** _ **10** _	**Error %**
Null model (inc. subject)	0.200	0.011	0.046	1.000	
RM Factor + Condition + RM * Condition	0.200	0.945	69.335	83.962	2.528
RM Factor	0.200	0.031	0.126	2.713	6.920
RM Factor + Condition	0.200	0.009	0.038	0.834	2.510
Condition	0.200	0.003	0.013	0.297	0.664
Note. All models include subject					
					
**Table 1b**					
Bayesian ANOVA					
Data Input: OD performance changes					
Model Comparison					
**Models**	**P(M)**	**P(MIdata)**	**BF** _ **m** _	**BF** _ **10** _	**error %**
Null model	0.500	0.010	0.010	1.00	
Condition	0.500	0.990	99.956	99.956	5.865e -4
					
**Table 1c**					
					
Bayesian ANOVA					
Post Hoc Comparison – Cond					
Models	**Prior Odds**	**Posterior Odds**	**BF** _ **10,u** _	**error %**	
Parietal vs Sham	0.587	18.103	30.818	2.505e -4	
Parietal vs hMT	0.587	11.304	19.243	8.095e 4	
Sham vs hMT	0.587	0.299	0.510	6.066e -4	

The Bayes factor analyses suggest that the significance of the p-value (and related effects sizes) found with conventional statistical analysis likely represent a real significant effect of parietal stimulation in the orientation discrimination task.

– There was still concern about the electric field models presented in the paper and their relationship with the activity in the stimulation sites that is driven by the tasks at hand. In consultation, one reviewer remarked, "The SimNIBS e-field models are presented in a raw form that in my experiences is only shown as a challenge getting the software to display the e-field on the cortical surface. I suggest that the authors run their e-field models in the ROAST toolbox. As presented, the e-fields are not really interpretable." The reviewers felt it would be important to compare whole-brain maps of task-related activity with accurate electric field models to determine whether the effects of stimulation are due to more effective targeting of task-active regions.

We have now run the e-field models using ROAST (see Figure 8 Methods section). We will respond to the second query by the reviewer (to compare whole-brain maps of task-related activity) more extensively below, to the specific question asked by reviewer 5. The connectivity changes we observe at rest are likely driven by stimulation and learning (Lewis et al., 2009), and the differences we report are stimulation site specific because the learning effect is selective for the parietal condition only.

– Several issues with clarity and claims made in the discussion need to be resolved. Please see the specific comments made by reviewers #4 and #5 below.

Thank you, we have now reviewed the manuscript and/or corrected and clarified.

Reviewer #4:N=10 / group is fairly small. Please report effect sizes. How was power calculated for the study?

We now have modified the manuscript to report effect sizes. We reported partial eta squared (η_p_^2^) for the ANOVAs and Cohen’s d for t-test. As we reported in the “transparent reporting” document (and copied here), an explicit power analysis was not performed. However, the sample size was based on previously published studies that used neurostimulation procedures coupled with learning and/or neuroimaging techniques and have a similar (or fewer) number of participants.

In particular, Helfrich and colleagues (Plos Biology, 2014) used a combination of high-density tACS and EEG testing 14 participants and used a very similar experimental procedure to ours, as they had two separate days and three stimulation conditions. Another study that used a combination of behavioral, neurostimulation and neuroimaging techniques (Saiote et al., 2013; PLOS ONE), investigated BOLD activity induced by high-frequency tRNS during a visuomotor task (10 subjects per condition) and found a behavioral improvement after hf-tRNS and cathodal tDCS and a reduction of learning-related brain activity.

In a different study that used the combination of learning and neurostimulation techniques, Snowball and colleagues (Snowball et al., 2013, *Current Biology: CB*, *23*(11), 987–992) tested 25 subjects altogether across 4 conditions. Another study conducted by Lewis and colleagues (Lewis et al., PNAS, 2009) investigated the role of spontaneous brain activity for brain dynamics of 14 participants. Other studies that used a combination of neurostimulation techniques and behavior and/or neuroimaging techniques and used a similar (or smaller) number of participants are reported in the transparent reporting document.

Unclear why data is represented inconsistently (i.e., post-stimulation scores at times and difference between FC scores pre- and post-test sessions at other times). Does post-stimulation refer to the change from baseline or just data from the post-session alone? The difference between FC scores pre- and post-test sessions is more informative and should be what is reported in the manuscript.

We apologize for the confusion. When we talk about “post-stimulation” we refer to data from the post-session alone, while when we talk about the difference between FC pre and post-test sessions we refer to the change in connectivity scores from baseline. The analyses on post-stimulation FC scores were performed to capture whether there was a difference in connectivity measures during the post-stimulation scanning session between stimulation conditions, given that there was no statistically significant difference when FC scores were compared between conditions in the pre-stimulation session. However, because we agree with the reviewer that it is important to compare how (and how much) connectivity changed between the two session, analysis on the difference in connectivity scores were also conducted and reported. As requested by the reviewer, we have now just reported the difference between the pre- and post-test FC scores. Moreover, we have removed one figure representing the post-stimulation FC, and replaced it with a Figure representing the difference in connectivity within the two networks (DVAN and DMN, Figure 6 in the manuscript).

Scatter plots should be removed.

We have removed the scatter plots as requested by the reviewer.

Reviewer #5:This study shows a very striking effect of parietal tRNS on object discrimination and corresponding large changes in functional connectivity. Such data are sure to be of interest to numerous researchers due to their practical significance. That said, I have statistical concerns and am left wanting more mechanistic insight. Ultimately, I think more data and analyses will be needed to make a convincing enough case to the readership of this outlet.1) The effect of parietal tRNS on object discrimination is truly massive. The tasks are titrated at the start to set performance at 50%. While performance remains the same following temporal and sham tRNS, parietal tRNS improves performance by 30%. Given this finding, I am rather convinced that parietal tRNS improves object discrimination. What I am not convinced of is that the effect size is as massive as is being depicted. Given that the main contribution of this study as it is current written is practical, having a reasonable estimate of effect size is important for applied work that will seek to build on this striking finding. However, given the small sample size (10 participants per group), and effect size inflation that occurs when conditionalizing on significance, especially in small samples, I suspect that the true effect size is substantially smaller than has been depicted. Unfortunately, I do not see a good way to get a better estimate of the true effect size without acquiring more data. The upshot is that new data would not require fMRI and may not even require control groups. A bare minimum would be to collect another group of 10 participants receiving parietal tRNS and estimate effect size based on their improvement. I think that such data collection would be a low cost effort, and the sample size is small enough to potentially be feasible amidst pandemic restrictions. Note that since the new sample would be used to estimate effect size, there need not be statistically significant improvement in the new sample to have utility.

We understand the request and rationale of the reviewer to collect additional data, and for this and other reasons, we decided to run further analyses. Specifically, we run Bayes factor analysis to estimate the amount of evidence present in the data. All details are reported above, in the first response to the Editor’s comments. We believe the additional analyses we conducted provide a clear evidence that the effect we found is not inflated.

2) These data beg the question of why parietal tRNS produces such a dramatic effect, but temporal tRNS does not. That is, one would like to have some mechanistic insight into why parietal tRNS is so effective, at least in the object discrimination task. The connectivity data do not, in their present form, address this question. They merely provide another metric of change without begetting further causal inference/insight. Fortunately, the authors have additional task fMRI data that could potential shed some light on the issue. For example, one might ask whether the extent to which a given task activates the stimulated areas is associated with changes in connectivity and/or behavior. Alternatively, the network position of the parietal target either generally or specific to a given task may be associated with its effective change in connectivity and/or behavior. Given that the authors have effects that are specific not only to target, but also differ as a function of task, I feel as though these data could be leveraged to provide substantial insights into what factors are critical for an effective intervention.

We agree with the reviewer’s suggestion that looking at task-related functional data might be interesting to better understand why parietal tRNS produces such a dramatic effect on behavior and on functional connectivity within the Dorsal and Ventral Attention network. It is indeed in our plan to carefully analyze and look at those data next in a follow-up manuscript. Those analyses will respond to a different set of questions; those indicated by the reviewer, and, in addition, whether the effect is driven by the type of task or by the effort required in switching between tasks and/or the “choice” of parietal functions to prioritize one task over the other. The wealth of information these analyses provide warrants a manuscript focused specifically on these questions which diverge from the current research question at hand. In the present study, we chose to observe functional connectivity changes at rest, activity that is not directly driven by the tasks, albeit we hypothesize it is the consequence of the stimulation and training sessions. Our choice was motivated by the fact that resting-state data have been shown to be particularly reliable and stable across subjects and across time (Damoiseaux et al., 2006), indicating that baseline activity of the brain exhibits consistent temporal and spatial dynamics that are also functionally relevant (e.g. dorsal and ventral attention networks have been found to be persistent in the absence of a task; Fox et al., 2006). Indeed, we found that resting-state connectivity within the attention network to be active at rest when measured in between attentional task runs and, importantly, to be significantly influenced by different stimulation protocols that are likely to have supported changes in behavior.

However, given the reviewer’s comment, we conducted a new set of analyses to investigate if the parietal regions we stimulated were also active during the tasks. In particular, we analyzed BOLD signal changes in response to the OD and the TOJ task in the pre-stimulation session, before any manipulation (e.g., stimulation and training) could potentially change the brain response to the tasks. Therefore, we included all subjects in the analysis (N=30) and we observed the brain areas significantly activated while performing the OD and the TOJ task. As visual stimuli were presented in rapid succession within each run, we analyzed BOLD activity using a deconvolution general lineal model (GLM) to separate the contribution of different events that could otherwise overlap. This deconvolution analysis included predictors for each volume (sampled at 2 sec), each task-condition, extending 20 sec following the onset of a trial. Statistical contrasts were estimated from the four volumes surround the peak of the hemodynamic response (4 sec post stimulus onset) for each voxel. In the first-level analysis, a GLM regression model was performed to fit data for each subject. Individual subject data was then combined into a deconvolution GLM in which β estimates on OD and TOJ trials were contrasted with estimates for baseline activity. This contrast identified brain areas associated with the OD task performance and TOJ performance, with significance assessed using a threshold false discovery rate of.05 (FDR corrected).

Figure 8 shows the results of a second-level deconvolution GLM analysis for significant positive BOLD response for the OD and the TOJ tasks versus rest (Figure 8, Panel A). Details of statistical maps with Talairach coordinates are reported in Table 2 and 3. Moreover, as suggested by the reviewer, we performed a simulation of the current distribution in the brain using ROAST (Huang et al., 2019) for the two stimulation conditions (Figure 8, Panel B for Parietal stimulation, and Panel C for hMT stimulation) and included them in the Figure to qualitatively assess the overlap between positive BOLD response and current distribution peaks over the cortex. As expected, the task-related BOLD activity for the OD and TOJ tasks peaked over the parietal lobes, which directly correspond to the hotspots where the peaks of the electrical current were localized during Parietal stimulation (Figure 8B). In contrast, current distribution during hMT stimulation peaked over occipital cortical regions. Interestingly, while the current distribution for Parietal stimulation remains relatively circumscribed to the posterior cortical areas, more anterior areas were significantly active during the tasks, as indicated by the BOLD response (Figure 8A). Thus, to directly answer the reviewer’s question, we conclude that the effects driven by Parietal stimulation are likely due to an effective targeting of selective task-active crucial nodes (e.g., IPS) that, as suggested by previous studies, are pivotal to task performance (Capotosto et al., 2013; Suzuki & Gottlieb, 2013; Corbetta & Shulman, 2002). However, we prefer to choose a more parsimonious interpretation of the results of the E-field models relative to the BOLD response for the following reasons. Despite the fact that the E-field models are being constantly updated, and they surely hold great promise for individualized tES modeling, their validation for some experimental condition including ours is still not sufficient. Recently, Callejón-leblic and Miranda (2021) acknowledge that the few in vivo data only show a moderate correlation with intracranial cortical recordings in humans (e.g. Opitz et al., 2018), and the results obtained using different modelling pipelines (Huang et al., 2019; Saturnino et al., 2019) are divergent (Puonti et al., 2020). Furthermore, an important limitation is the difficulty to predict whether a stimulation procedure will influence the site of stimulation only, or whether it will exert a network effect, as we found in our resting-state data. In addition, while functional data can be observed as a change across time and sessions (between pre- and post-stimulation and training), this parameter cannot be computed in the E-field model, as the outcome is based on fixed parameters such as electrodes’ size, location and intensity of stimulation but not on stimulation duration nor the number of sessions, which we believe to be a crucial factor in our study. Since the time variable cannot be included in the simulation, it becomes difficult to interpret the E-field model in a multi-day stimulation protocol, and it remains to be further investigated/implemented in the future.

Relatedly, the authors offer conclusions in the discussion regarding what constitutes a successful intervention, but they are either unsubstantiated by the present data, or unclear. For example, they state on page 20, lines 439-441:"It is the combination of a challenging task with concurrent brain stimulation over the relevant cortical circuits that facilitates fast learning, a paradigm amenable to promote plasticity."

We have now edited the text and added some details to the Discussion section.

Similar sentiments are echoed on page 20, lines 451-453. Absent examination of task data, the relevant cortical circuits for the studied tasks are unclear. Consider that in Figure 9, the temporal targeted tRNS seems to induce the most stimulation in occipital areas. I would expect occipital cortex to be extremely relevant for the studied tasks, yet stimulation over those areas did not improve performance. Moreover, the importance of stimulation concurrent with tasks cannot be concluded from these data since there were no conditions that had no tasks. The differential performance across the studied tasks may potentially be leveraged here, but at the moment, this conclusion does not follow from the data. Since the authors have data that can help speak to some of these issues, it seems pertinent to use them.

We agree with the reviewer, and we have now changed the text and removed the statement about the importance of delivering stimulation concurrently with the task. However, in a preliminary dataset collected prior to the current study, we also had a condition with tRNS alone and without a concurrent task (whilst we used the same behavioral tasks at pre-stimulation baseline, and during the post-stimulation session). Those data are not published yet, but they have been presented at a meeting in December 2020 (Transcranial Brain Stimulation in Cognitive Neuroscience: https://event.unitn.it/tbs-cnw/#home, “Functional response of attention-related cortical networks to multi-sessions tRNS” Pergher, V., Contò, F. & Battelli L.). These unpublished preliminary data indicate that tRNS alone is not sufficient to boost learning with the magnitude we found when tRNS and behavioral training are run simultaneously.

[Editors' note: further revisions were suggested prior to acceptance, as described below.]

The manuscript has been greatly improved but there are some remaining issues that need to be addressed, as outlined below. Though these revisions are essential, the manuscript will not be sent out for another round of external reviews and can be handled editorially.1. While the Bayesian analyses are nice, they do not address the point that the effect sizes are likely to be inflated given the sample size. The issue is that when sample size is small and reporting of results is conditioned on statistical significance (e.g. p< 0.05) , effect size will be inflated. This needs to be acknowledged in the discussion. See, for example, the reference below for more discussion of inflated effect sizes in studies with small samples.Yarkoni, T. (2009). Big correlations in little studies: Inflated fMRI correlations reflect low statistical power-Commentary on Vul et al. (2009). Perspectives on psychological science, 4(3), 294-298.

We have now addressed the reviewer’s point in the Discussion section by pointing out this potential limitation in our study.

2. Please report the analyses showing that the number of frames censored were similar across sessions in the manuscript.

We have now reported the analysis in the manuscript, in the section: “Stimulation and learning-dependent changes in functional connectivity”.

3. The task-based univariate results are quite illuminating. It would seem pertinent to mention in the manuscript that both tasks activate IPS in the discussion of why effects were observed for the OD task but not the TOJ task.

We agree, this is a very interesting point and we have now added a paragraph in the discussion (bottom of page 20) where we offer an interpretation of why we found this differential effect in behavior that is not evident in the task-based response at baseline. As we mentioned in our previous response, this clearly generates a set of interesting hypotheses that we plan to analyze next in a follow-up manuscript. We have also added the task-based functional connectivity analysis at the end of the methods section and modified Figure 8, where we now report the task-based univariate results (8A) and the current distribution results from ROAST (8B and C).